# Endothelial Cell Dysfunction and Nonalcoholic Fatty Liver Disease (NAFLD): A Concise Review

**DOI:** 10.3390/cells11162511

**Published:** 2022-08-12

**Authors:** Narjes Nasiri-Ansari, Theodoros Androutsakos, Christina-Maria Flessa, Ioannis Kyrou, Gerasimos Siasos, Harpal S. Randeva, Eva Kassi, Athanasios G. Papavassiliou

**Affiliations:** 1Department of Biological Chemistry, Medical School, National and Kapodistrian University of Athens, 11527 Athens, Greece; 2Department of Pathophysiology, Medical School, National and Kapodistrian University of Athens, 11527 Athens, Greece; 3Warwickshire Institute for the Study of Diabetes, Endocrinology and Metabolism (WISDEM), University Hospitals Coventry and Warwickshire NHS Trust, Coventry CV2 2DX, UK; 4Warwick Medical School, University of Warwick, Coventry CV4 7AL, UK; 5Laboratory of Dietetics and Quality of Life, Department of Food Science and Human Nutrition, School of Food and Nutritional Sciences, Agricultural University of Athens, 11855 Athens, Greece; 6Third Department of Cardiology, ‘Sotiria’ Thoracic Diseases General Hospital, Medical School, National and Kapodistrian University of Athens, 11527 Athens, Greece; 7Endocrine Unit, 1st Department of Propaedeutic Internal Medicine, ‘Laiko’ General Hospital, Medical School, National and Kapodistrian University of Athens, 11527 Athens, Greece

**Keywords:** vascular endothelial cells, sinusoidal endothelial cells, endothelial dysfunction, NAFLD, LSECs, CVD, inflammation

## Abstract

Nonalcoholic fatty liver disease (NAFLD) is one of the most common liver diseases worldwide. It is strongly associated with obesity, type 2 diabetes (T2DM), and other metabolic syndrome features. Reflecting the underlying pathogenesis and the cardiometabolic disorders associated with NAFLD, the term metabolic (dysfunction)-associated fatty liver disease (MAFLD) has recently been proposed. Indeed, over the past few years, growing evidence supports a strong correlation between NAFLD and increased cardiovascular disease (CVD) risk, independent of the presence of diabetes, hypertension, and obesity. This implies that NAFLD may also be directly involved in the pathogenesis of CVD. Notably, liver sinusoidal endothelial cell (LSEC) dysfunction appears to be implicated in the progression of NAFLD via numerous mechanisms, including the regulation of the inflammatory process, hepatic stellate activation, augmented vascular resistance, and the distortion of microcirculation, resulting in the progression of NAFLD. Vice versa, the liver secretes inflammatory molecules that are considered pro-atherogenic and may contribute to vascular endothelial dysfunction, resulting in atherosclerosis and CVD. In this review, we provide current evidence supporting the role of endothelial cell dysfunction in the pathogenesis of NAFLD and NAFLD-associated atherosclerosis. Endothelial cells could thus represent a “golden target” for the development of new treatment strategies for NAFLD and its comorbid CVD.

## 1. Introduction

Nonalcoholic fatty liver disease (NAFLD) is the leading cause of chronic liver disease worldwide [1,2]. NAFLD is closely related to the features of metabolic syndrome, with insulin resistance (IR) being the key pathogenic feature [3]. Hyperinsulinemia, hyperglycemia, lipotoxicity, and altered adipocytokine secretion can activate deleterious processes such as inflammation, oxidative stress, endoplasmic reticulum (ER) stress, and apoptosis, which lead to the development of NAFLD, demonstrating its multifactorial etiology. Despite its high prevalence, there are currently no US Food and Drug Administration (FDA)-approved treatments for NAFLD [4].

Endothelial cells cover the inner surface of arteries, veins, and capillaries and form a barrier between the blood and tissues [5]. Due to the privileged position and intimate contact with the blood stream, endothelial cells are the first line facing various circulating stimuli produced by neighboring cells or distant sites [6,7,8].

Of note, intrahepatic vascular alteration appears to contribute greatly to the NAFLD pathogenesis process. Liver sinusoidal endothelial cells (LSECs) are a very distinct type of endothelial cells that form the wall of the hepatic sinusoids. LSECs govern the regulation of the hepatic microenvironment and act as the hepatic first defense barrier. In addition, LSECs participate in the regulation of the hepatic cellular response to various injuries through the regulation of neighboring cells’ functions’ such as hepatic stellate cells (HSC) and immune cells [6,7,8]. The alteration of the hepatic endothelium contributes to the development of NAFLD and liver fibrosis [2,7,8,9]. New evidence identifies LSEC dysfunction as the main characteristic or early event in the development of liver pathology in NAFLD, contributing to impaired hepatic lipid uptake and metabolism, disturbed macromolecules and metabolite transport, angiogenesis, intrahepatic inflammation, hepatocellular damage, and finally the impairment of hepatic blood flow with intrahepatic resistance [2,7,8,9,10,11].

Vice versa, NAFLD occurrence leads to a higher risk of vascular endothelial dysfunction and atherosclerosis progression, independent of the occurrence of metabolic syndrome and its components [12].

Considering the involvement of endothelial cells as important regulators of NAFLD progression and its co-morbidities such as CVD, this review is focused on addressing the role of LSEC dysfunction in the pathogenesis of NAFLD. Inversely, the effect of hepatic secreted molecules on the initiation of vascular endothelial dysfunction and atherosclerosis is also discussed. Finally, data regarding the role of endothelial cells as a target for the development of new treatment strategies for NAFLD and its comorbidities are also provided.

## 2. NAFLD Epidemiology and Pathogenesis

Nonalcoholic fatty liver disease (NAFLD) includes a spectrum of hepatic disorders, ranging from liver fat deposition in more than 5% of hepatocytes (steatosis—nonalcoholic fatty liver (NAFL)) to necroinflammation and fibrosis (nonalcoholic steatohepatitis (NASH)), which can progress into NASH-cirrhosis, and eventually—albeit rarely—to hepatocellular carcinoma [13,14]. NAFLD shows an increasing prevalence alongside the growing epidemic of obesity, reaching 25% worldwide and ranging from 13% in Africa to 42% in South-East Asia [15,16]. The most common predisposing factors for NAFLD are male sex, age > 50 years, hyperlipidemia, obesity, insulin resistance and T2DM, and a lack of physical exercise as well as genetic polymorphisms (e.g., patatin-like phospholipase domain-containing 3 (PNPLA3) I148M polymorphism) [9,17,18,19]. Reflecting the underlying pathogenesis and the cardiometabolic disorders associated with NAFLD, the term metabolic (dysfunction)-associated fatty liver disease (MAFLD) has recently been proposed [4].

Even though NAFLD has distinct predisposing factors, its exact pathogenesis is currently unrevealed. For years, the “two-hit” theory was the prevailing one; according to this theory, the pathophysiology of NAFLD consisted of a first “hit” representing the stage of simple steatosis with lipid accumulation and insulin resistance [20], followed by a second “hit”, leading to oxidative and endoplasmic reticulum (ER) stress, leading to the development and progression of hepatic inflammation and fibrosis. Nowadays, the “two-hit” theory is replaced by the “multiple parallel-hit” model [21] that attempts to explain the complex pathogenesis of liver inflammation and consequent fibrosis. According to this theory, different amalgamations of numerous (epi)genetic and environmental factors (such as specific genetic polymorphisms and epigenetic modifications [22], features of metabolic syndrome [23,24,25,26], lipotoxicity [27,28], dysbiosis of the gut microbiota [28], dysregulation of autophagy and mitochondrial function [29,30,31], ER stress [30,32], and hepatocyte homeostasis and death [33,34] as well as inflammatory and fibrotic responses [35,36]), representing “hits”, dynamically interplay with each other, leading to NAFLD development and progression. Notably, when the hepatic capacity to handle the primary metabolic energy substrates is overwhelmed, toxic lipid species accumulate in the liver, leading to hepatocyte dysfunction and apoptosis, along with metabolically triggered inflammation and subsequent fibrosis [37].

## 3. Endothelial Cells in the Pathogenesis of NAFLD

Apart from the risk of liver fibrosis/cirrhosis, patients with NAFLD have an increased risk of all-cause mortality, especially due to cardiovascular disease (CVD) [38,39,40]. This risk is attributed to the common predisposing factors for both NAFLD and CVD, with the endothelium emerging as a key player [41,42,43,44]. Recent data indicate that both vascular endothelium dysfunction and LSEC dysfunction play particularly significant roles not only in the pathogenesis and progression of NAFLD but also in the interplay between CVDs and NAFLD [42,45].

The vascular endothelium participates in the regulation of various physiological and pathophysiological processes, such as inflammation, angiogenesis, vascular tone, platelet function, and metabolic homeostasis [2,46].

On the other hand, LSECs, a very specialized and phenotypically differentiated endothelium with a unique anatomical location and structure, are not only responsible for controlling material exchanges between the liver parenchyma and the circulation, but they also maintain the anti-inflammatory, anti-thrombotic and, anti-fibrotic milieu within the liver parenchyma [8,11,47]. Interestingly, LSECs balance the fibrosis and regeneration process in liver through the secretion of angiocrine factors in response to hepatic injury [48]. LSECs are the key regulators of hepatic homeostasis, blood flow, endocytic capacity, and the inflammatory response during the whole course of NAFLD [2].

### 3.1. LSECs and Their Role in the Regulation of Blood Flow and Hepatic Microcirculation

LSECs, which comprise approximately 20% of the total number of hepatic cells, belong to the hepatic nonparenchymal group of cells that are placed at an interface between the hepatic parenchyma and the blood of the hepatic artery and portal vein [49,50]. LSECs differ from other endothelial cells of the body due to their unique morphological structure that is characterized by the presence of small pores called fenestrae and the lack of a basement membrane as well as a diaphragm [51]. While LSECs represent a barrier for macromolecule transport, albeit not in the same way as vascular endothelial cells in other organs, they are regarded as the most permeable endothelial cells in mammals [49]. Specifically, LSEC fenestrae connect the Disse space to the sinusoidal side and permit the entrance of lipoproteins, chylomicron remnants, and other macromolecules from the circulating blood to the Disse space and their utilization by hepatocytes [2].

On the other hand, hepatic endothelial cells are in charge of blood flow regulation, as are the endothelial cells of other tissues [8,51]. Various studies indicate that disorders of hepatic microcirculation, such as decreased hepatic blood flow, may play a critical role in the pathogenesis and progression of chronic liver diseases, including NAFLD [49,52,53] (Figure 1).

This image was derived from the free medical site http://smart.servier.com/ (accessed on 1 June 2022) by Servier, licensed under a Creative Commons Attribution 3.0 Unported license.Abbreviations: CCL: C-C motif chemokine ligand; CCR: C-C motif chemokine receptor; CXCL: C-X-C motif chemokine ligand; NO; nitric oxide; ICAM-1: Intercellular adhesion molecule-1; LSEC: Liver sinusoidal endothelial cell; TNF-α: Tumor necrosis factor-α; VAP-1: Vascular adhesion protein-1; VCAM-1: Vascular cell adhesion molecule-1; N: Neutrophils; ROS: Reactive oxygen species; VEGF: Vascular endothelial growth factor; VEGFR: Vascular endothelial growth factor receptor; EV: Extracellular vesicles; KLF2: Kruppel-like factor 2; M: Monocytes; KC: Kupffer cells; IR: Insulin resistance; HSCs: Hepatic stellate cells; SR: Scavenger receptor; MR: Mannose receptor; FcγRIIb2: Fc gamma receptor IIb. 

Hepatic microcirculation impairment and the reduction in hepatic flow are mainly the result of structural and functional changes in LSECs. However, other factors, such as the lipid-induced enlargement of parenchymal cells, the secretion of vasoactive factor nitric oxide (NO) by Kupffer cells (KC), increased oxidative stress as indicated by increased reactive oxygen species (ROS) production, the activation and contraction of HSCs, and the activation and secretion of pro-inflammatory and profibrogenic cytokines, along with the deposition of extracellular matrix (ECM) proteins into the Disse space, have been identified as potent contributors [42,49,54,55]. Indeed, in a model of diet-induced hepatic steatosis in Sprague Dawley rats, NO was found to play a role in the modulation of hepatic microcirculation, part of which is disrupted due to the sinusoidal compression caused by the enlargement of hepatocytes [56].

According to in vivo and in vitro data, elevated oxidative stress in LSECs has been directly linked to reduced microvascular blood flow [57,58]. Sun et al., using genetically obese Zucker rats, showed that changes in the sinusoidal blood flow may contribute to worsening hepatic injury through oxygen deprivation in centrilobular regions as well as the modulation of nutrient exchange between hepatocytes and the vasculature [59]. Of note, the extension of hepatic steatosis observed during NAFLD progression has been inversely related to hepatic microcirculation blood flow (HMBF) [53,60]. A recent study by da Silva Pereira et al., revealed that HMBF in high-fat diet (HFD)-fed Wistar rats is reversibly correlated with body weight (BW), fasting blood glucose (FBG), and visceral adipose tissue (VAT), indicating the involvement of metabolic parameters in the microcirculation impairment [57]. Male Wistar rats under HFD for 20 weeks demonstrated a 31% decrease in HMBF [57]. A caloric restriction through changing from a HFD to a chew diet for 8 weeks partially reversed the HMBF reduction, while the combination of diet intervention for 8 weeks along with pyridoxamine (PM) supplementation resulted in complete HMBF prevention. In fact, these data substantiated that lifestyle modifications along with PM supplementation may restore endothelial function and normal HMBF, serving as an effective treatment for such hepatic complications [57].

The evaluation of hepatic microvascular alteration in steatotic livers of Zucker rats showed an abnormal microcirculation manifested by a reduced sinusoidal density compared to the control group [59]. The assessment of liver blood flow and LSEC function in ob/ob mice with severe liver steatosis showed impaired liver blood flow and sinusoidal perfusion, which was further worsened after hepatic ischemia injury [61]. The impairment of microcirculation in these ob/ob mice resulted in the induction of ischemic necrosis compared to the lean animals [61]. Hyperphagic mice lacking a functional *Alms1* gene (Foz/Foz) can be used as a genetic/dietary model of both simple steatosis (under chew diet) and NASH (under HF diet) [62]. In this model, a substantial alteration in blood flow was observed in the livers of mice with NASH or simple steatosis due to the sinusoidal structural alteration. In detail, hepatic lipid accumulation caused hepatic parenchymal cell enlargement, leading to the parenchymal cell plate widening and a narrowing of the sinusoid lumen [62]. Confirming these findings, another study in diet-induced NAFLD rats showed that the hepatic parenchymal cell enlargement and swelling due to lipid accumulation caused a reduction in sinusoidal perfusion; these alterations resulted in a distortion of the sinusoidal endothelial cell lumen and a reduction in intrasinusoidal volume, leading to LSEC architectural changes and the impairment of tissue perfusion [54]. In 10% lipogenic MCD-diet-induced NAFLD mice, the sinusoidal perfusion was found to be impaired due to narrowed sinusoidal lumens along with induced perivascular fibrosis; of note, the duration of the dietary intervention was positively related to the degree of sinusoidal perfusion [63].

In line with the aforementioned study, Seifalian et al., using another diet-induced NAFLD rat model, confirmed that decreased blood perfusion in the microcirculation is strongly related with the severity of steatosis and lipid accumulation in hepatocytes [64]. Furthermore, reduced hepatic vascular density and blood flow, along with pronounced HSC activation and increased leukocyte recruitment in the sinusoidal and postsinusoidal venules have been observed in the hepatic microcirculation of mice with NASH [58].

As previously mentioned, LSECs regulate blood flow through the secretion of vasoactive substances in the liver, namely, NO and endothelin-1 (ET-1). It is well-known that both of these molecules counter-regulate vascular tone, with NO promoting vascular dilation, while ET-1 induces the contraction of blood vessels. LSECs regulate the expression of both NO and ET-1 through an endothelial-specific transcription factor known as Kruppel-like factor 2 (KLF2) [2,65,66]. In particular, reduced eNOS activity and decreased NO production have been linked to reduced KLF2 expression [67,68].

Increased portal pressure (PP) and decreased endothelium-dependent vasodilation were observed in perfused livers of CafD (65% fat, mostly saturated)-fed Wistar Kyoto rats, as compared to the control group [8]. The observed increased hepatic vascular resistance in CafD rats was associated with reduced Akt-dependent eNOS phosphorylation and eNOS activity. The authors concluded that in this specific rat model of metabolic syndrome with NAFLD features liver endothelial dysfunction occurs prior to hepatic inflammation or the development of fibrosis [8]. Of note, it has been shown that liver eNOS expression is negligible outside endothelial cells. Thus, changes in hepatic eNOS phosphorylation represent changes in the liver endothelium only [8,69,70].

While small amounts of NO generated by eNOS are believed to have hepatoprotective effects, iNOS expression is associated with the development and maintenance of NAFLD. In fact, iNOS expression is absent in resting cells, while it is induced during inflammation [71,72]. iNOS mRNA levels were elevated after the stimulation of LESCs with IFN-γ, while the incubation of cells with interleukin-1β (IL-1β) and tumor necrosis factor-α (TNF-α) had no effect on iNOS levels [73]. The overexpression of iNOS in the liver tissues of an NAFLD animal model leads to induced oxidative stress and a pro-inflammatory response followed by hepatic microcirculation dysfunction [74,75].

NO bioavailability can also be altered by the oxidative stress generated during the progression of NAFLD/NASH, while antioxidants and redox environments are crucial for the maintenance of microcirculation during compromised liver perfusion [76,77]. IR, the main feature of cardiometabolic diseases, induces oxidative stress and iNOS levels, leading to the development of endothelial dysfunction. A recent work by Gonzalez-Paredes et al., confirmed the occurrence of LSEC dysfunction in Sprague Dawley rats after 6 weeks of HFD, as indicated by decreased NO activity and increased oxidative stress [78]. HFD-induced oxidative stress (indicated by increased hepatic MDA and 3-nitrotyrosination (3-NT) and lowered p-eNOS levels) in these rats led to the impairment of endothelium-dependent relaxation compared to the control diet. Of note, hepatic endothelial dysfunction was improved after pre-treatment with an antioxidant agent, highlighting the potential of antioxidant therapy at early stages of LSEC dysfunction, even prior to the activation of pro-inflammatory and profibrogenic pathways [78]. Circulating lipids were found to induce oxidative stress in LSECs, while this oxidative stress contributes to hepatocyte injury, resulting in NASH [79]. Indeed, the treatment of primary murine cultured LSECs with palmitic acid (PA) upregulated the expression of the NOX1 isoform of NADPH oxidase, an enzyme implicated in ROS production [80]. Furthermore, NOX1 was also upregulated in the liver of NASH patients and mice fed a high-fat and high-cholesterol (HFC) diet for 8 weeks, while mice deficient in NOX1 displayed decreased levels of serum alanine aminotransferase (ALT) and hepatic cleaved caspase-3 compared to wild-type littermates when fed the HFC diet [80].

Taking into account all the above, it appears that changes in hepatic blood circulation during the early stage of steatosis progression to fibrosis are regulated by LSECs. LSECs play crucial roles in sensing and regulating portal pressure, hepatic vascular resistance, and finally hepatic microvascular blood flow via various mechanisms that mainly implicate the key vasodilator factor, eNOS/NO.

### 3.2. Capillarization of LSECs in NAFLD

The progression of simple nonalcoholic steatosis to steatohepatitis and fibrosing steatohepatitis is closely related to the initiation of sinusoid capillarization. Through this structural transformation, the progressive loss of fenestrae in the LSECs is accompanied by the development of a basal lamina and collagen deposition in the Disse space, which in turn leads to sinusoidal lumen narrowing and distortion and a consequent microvascular blood flow reduction [54,63].

The capillarization of LSECs promotes the development of steatosis in NAFLD through the prevention of very-low-density lipoprotein (VLDL) release from hepatocytes into the sinusoidal cavity and the consequent preservation of hepatic cholesterol and triglycerides in the liver [79]. Furthermore, it induces hepatic *de novo lipogenesis* and VLDL synthesis through the impairment of chylomicron remnant entry into hepatocytes, leading to increased hepatic damage and steatosis/fibrosis [79,81].

The capillarization of LSECs occurs in the very early phase of NAFLD, even prior to steatosis establishment [79]. Indeed, LSEC defenestration begins after 1 week of choline-deficient L-amino acid-defined (CDAA) diet administration in mice [82], and LSEC morphology is damaged after 3 weeks of HFD feeding in rats [83]. Capillarization then leads to liver steatosis, as shown in mice deficient in plasmalemma vesicle-associated protein (PLVAP), an endothelial-specific integral membrane glycoprotein that has been identified to be a component of endothelial fenestrae [81]. The LSECs of these mice exhibit a significant reduction in the number of fenestrations, which is associated with a decrease in the transport of macromolecules from the sinusoidal lumen into the Disse space, leading to the development of extensive multivesicular steatosis, followed by steatohepatitis and fibrosis [81]. LSEC capillarization occurs before KC activation and is permissive of hepatic stellate cell activation and the progression of inflammation and fibrosis [8,79,84].

Leukocyte cell-derived chemotaxin 2 (LECT2), a functional ligand of Tie1 is a new hepatokine expressed by hepatocytes and LSECs [85,86] as well as vascular endothelial cells [87]. Upon binding to the Tie1 receptor, LECT2 facilitates Tie2/Tie2 homodimerization, activates the PPAR pathway, and inhibits tube formations and neo-angiogenesis. The overexpression of LECT2 inhibits portal angiogenesis and promotes sinusoid capillarization, leading to a worsening of hepatic fibrosis. Interestingly, the hepatic fibrosis changes were reversed in LECT2-KO mice, suggesting the LECT2-Tie1 signaling pathway as a potential target for liver fibrosis treatment [88]. The endothelial capillarization is mostly distinguished by the surface expression of CD31. LSEC capillarization and fibrosis were increased in the fibrotic liver of HFD-fed mice, as indicated by the increased expression of CD31 and Col1a1, respectively [89]. Of note, in contrast with the aforementioned data, CD31 was found to be highly expressed in LSECs obtained from both the control group and vaporized carbon tetrachloride (CCl4)-induced cirrhosis mice, regardless of the presence of cirrhosis, while the expression of CD34 was significantly higher only in the cirrhotic livers. This finding indicates that CD34 may represent a better marker for the detection of LSEC capillarization in cirrhotic livers [90,91].

In parallel with LSEC capillarization, bone-marrow-derived endothelial progenitor cells (BM-EPCs) are increased during chronic liver diseases [92,93]. LSEC capillarization induces hypoxia due to an elevated resistance to blood flow and oxygen delivery from the sinusoids to the parenchyma, leading to increased HSC activation and the expression of VEGF, angiopoietins, and their receptors [93,94,95,96]. A paracrine crosstalk between BM-EPCs and LSECs via VEGF and PDGF was previously reported. In particular, a study by Kaur et al., showed that the interaction between circulating BM-EPCs and resident LSECs enhances angiogenic functions via the induction of paracrine mediators such as VEGF and PDGF-BB. The same study reported induced circulating BM-EPCs levels in patients with cirrhosis compared to controls [97]. Another study from the same research group indicated that there is a substantial positive correlation between the abundance of BM-EPCs and fibrosis during the early stage of liver injury in mice (after 4 weeks of CCL4 treatment), while after 8 weeks of CCL4 treatment, EPC levels returned back to basal levels, most likely due to the lack of demand for hepatic tissue regeneration, indicating the potential role of BM-EPs in the early stage of liver fibrosis [93].

Another study by Liu et al., demonstrated that an intraperitoneal injection of cultured EPCs into rats under CCL4 treatment for 8 weeks exerted a hepatoprotective effect, as indicated by reduced ALT and AST levels and decreased liver fibrogenesis [98,99]. In line with these findings, the injection of cells derived from a high-density (HD) culture of rat bone marrow cells enriched in BM-EPCs to the CCL4-treated rats improved both the biochemical and fibrotic markers of liver injury 4 weeks post-transplantation. Importantly, the transplanted EPCs were not differentiated into either hepatocytes or endothelial cells, confirming that the BM-EPCs exert these beneficial effects, most likely by acting on surrounding cells rather than their direct interaction [93,100]. An increased number of BM-EPCs in NAFLD patients was considered to be a compensatory mechanism against the endothelial injury observed during NAFLD progression, and it was proportionally associated with the degree of liver steatosis [96,101]. On the contrary, a reduced number and function of circulating BM-EPCs in patients with NAFLD was reported by Chiang et al. This study revealed that the circulating BM-EPC population can be used as an independent reverse predictor of NAFLD [102].

### 3.3. LSECs in the Regulation of Inflammation in NAFLD

Endothelial dysfunction and changes in blood flow found in the liver microcirculation can be explained, at least in part, through increased inflammatory mediators, such as TNF-α, IL-1β, and chemokines. Inflammatory signals can trigger the expression of adhesion molecules, including vascular cell adhesion molecule 1 (VCAM-1), intercellular adhesion molecule 1 (ICAM-1), vascular adhesion protein-1 (VAP-1), and E-selectin from endothelium, leading to the recruitment of extravasation leukocytes and macrophages. An increased expression of VCAM-1 promotes the development of a firm adhesion between leukocytes and endothelium, leading to the formation of inflammatory foci and the activation of the inflammatory response [103].

When NAFLD progresses to NASH, the LSECs display a pro-inflammatory phenotype characterized by the surface overexpression of adhesion molecules such as ICAM-1, VCAM-1, and VAP-1 (AOC3) and the production of pro-inflammatory molecules, including TNF-α, IL-6, IL-1, and MCP1 (CCL2), as observed in experiments in mouse models of NASH [95,104]. The monocytes that adhered to the LSECs and were trapped in the sinusoids play a pivotal role in the initiation and progression of NAFLD [8,49,54].

Increased serum and tissue levels of adhesion molecules have been reported in both human and animal studies [8,105,106]. The specific role of neutrophils in the progression of NAFLD has attracted a great deal of research interest. Data show that neutrophils can release various compounds such as myeloperoxidase and elastase, which promote liver steatosis and a worsening of the inflammatory state, leading to liver damage. Notably, neutrophils were found to upregulate the expression of LSEC adhesion molecules, stimulating both endothelial cell and KC activation and the further recruitment of both monocytes and bone-marrow-derived macrophages (BMMs) [35,107]. BMM recruitment is also an important step in chronic liver inflammation [2]. LSEC dysfunction facilitates the recruitment and activation of macrophages (both resident KCs and BMMs) in the CDAA-diet-induced NASH mice model through—among others—the release of the C-C motif chemokine receptor (CCR) ligand, known as CC2. Both CC2 and CCR2 are expressed in monocytes and macrophages and are critical molecules for the recruitment of macrophages to the site of inflammation. Of note, a pharmaceutical inhibition of CCR2 prevented the infiltration of Ly6C-positive macrophages and promoted the alleviation of hepatic inflammation and fibrosis [2,108].

CXCR chemokine receptors and their CXC ligands (CXCLs) regulate the migration and homing of inflammatory cells to the liver [109]. Aberrant expression of CXCR4 has been observed in NAFLD. The binding of CXCR4 to its ligand, CXCL12, regulates cell localization, chemotaxis, activation, migration, proliferation, and differentiation [109,110]. CXCL12, also known as stromal-cell-derived factor 1α (SDF-1α), is extensively produced by LSECs and induces HSC migration during chronic liver injury [110]. Increased CXCR4 and CXCL12 protein levels, along with aberrant CD4+ T-cell responses to CXCL12, have been observed during the progression of NASH [110,111]. The hepatic recruitment of the CD4+ T-cell population is eased by LSECs through the increased peri-vascular expression of CXCL12 and the activation of CXCL12-CXCR4-dependent intracellular transport mechanisms [112]. CXCL12-CXCR4 axis activation induced HSC proliferation and increased the production of collagen I in a CCL4-induced hepatic fibrosis mouse model [113].

LSECs and KCs are hepatic antigen-presenting cells that are domestic to the liver sinusoidal lumen. Antigen presentation by LSECs to naive CD4+ and CD8+ T cells is upregulated by inflammatory stimuli and induces T-cell differentiation towards the regulatory phenotype (Treg) through the activation of TGF-β and/or Notch-dependent signaling mechanisms [114,115,116]. In turn, Tregs induce fibrogenesis by increasing the expression of both CD8+ and CD4+ T cells as well as the activation of Th17 cells [117]. Under physiological conditions, LSEC-driven antigen presentation to CD8 T cells mediates naïve CD8+ T-cell tolerance, while in the presence of high levels of antigen, this LSEC response is abrogated [114,115]. Additionally, LSECs are able to activate the naive CD4+ T cells and induce the expression of inflammatory cytokines by these cells. A study by Knolle et al., showed that the antigen of purified murine (female 12–16-week-old BALB/c mice) LSECs can efficiently activate CD4+ T cells, as indicated by the induced expression of IL-10, IL-4, and IFN-γ cytokines [115,118].

LSECs also express pattern recognition receptors such as stabilins and toll-like receptors (TLR 1-9). LSECs, in response to TLR ligand stimulation, except for that of TLR5, activate inflammasome and inflammatory signaling [116]. In more detail, LSECs produce either TNFα in response to the TLR1, 2, 4, 6, 9 ligands or TNFα, IL-6, and interferon (IFN) in response to TLR3 ligands [119]. Importantly, the above-mentioned LSEC response to TLRs through the secretion of different types of cytokines is cell-specific and was not observed in the KCs isolated from the same mice [120]. In NASH, the secretion of cytokines by LSECs leads to the release of inflammatory mediators and therefore facilitates the progression of disease [121]. Some inflammatory-related signals in LSECs are mediated through the interaction between the endocytosis receptors expressed on the LSECs and TLRs [2]. The TLR9 expressed by LSECs can take up the bacterial DNA mimic CpG oligonucleotides and activate endocytosis through a scavenger receptor, leading to the secretion of inflammatory cytokines such as interleukin (IL)-1β and IL-6 [122]. Accordingly, NAFLD activity scores, serum ALT levels, inflammatory cytokine expression, and hepatic TGF-β and collagen I expression were reduced in TLR4 KO mice under an MD diet compared to WT mice fed MD [123]. Of note, a study by Seki et al., demonstrated that fibrogenesis was significantly reduced after the treatment of TLR4 mutant mice with CCL4 or TAA [124].

The TLR4-mediated inflammatory signal is regulated by runt-related transcription factor 1 (RUNX1) [125]. RUNX1 is an lncRNA involved in the regulation of oxidative-stress-induced angiogenesis and inflammation during NAFLD [125,126]. Increased expression of RUNX1 has been positively correlated with steatosis, fibrosis, and the degree of hepatic inflammation as well as NASH activity score in NAFLD patients [125,127]. RUNX1 expression has been found to be upregulated in both diet- and CCL4-induced NASH mouse models, leading to HSC activation and the progression of NASH [128]. The expression of RUNX1 was increased in the LSECs of mice under an MD diet compared to control. An LSEC-specific RUNX1 silencing of mice under an MD diet resulted in reduced expression of ICAM-1, VCAM-1, and the infiltration of immune cells in NASH [126]. RUNX1 elevated the expression of angiogenic and chemotactic factors and adhesion molecules in HUVEC cells, while these EC properties were abolished after RUNX1 silencing, indicating the involvement of RUNX1 in enhancing inflammation and disease severity in NASH [127].

However, not all data point towards an unfavorable role of LSECs in the liver inflammatory process. A recent study demonstrated that at the early stage of NAFLD development LSECs exert anti-inflammatory effects through the suppression of leukocyte recruitment into hepatic sinusoids; this effect is mediated by reducing the expression of CCL2, CXCL10, and CXCL16. Indeed, decreased CXCL10 and CXCL16 expression after short-term exposure of both human and murine LSECs to FFA can lessen monocyte recruitment and inflammation [129]. The anti-inflammatory response seems to be mediated by the activation of MAPK signaling [108,129]. A research study by Neumann et al., showed that LSECs induced the expression of the anti-inflammatory cytokine IL-10 in developing Th1 cells [130]. A blockage of IL-10 signaling in vivo inhibited the immunosuppressive activity of LSEC-stimulated Th1. Moreover, LSECs induced the expression of Notch target genes hes-1 and deltex-1 in Th1 cells, leading to increased IL-10 expression and the activation of an anti-inflammatory response [130]. On the other hand, Notch signaling is also able to provoke LSEC dedifferentiation by regulating eNOS/sGC and Delta-like ligand 4 (DLL4) upregulation [131,132].

These data indicate that, while LSECs show an anti-inflammatory profile as the first line of defense at the initiation of NAFLD, this profile shifts towards a pro-inflammatory function, promoting the expression of adhesion molecules and the activation of HSCs and KCs, during the progression of NAFLD (Figure 2).

Defective autophagy has been observed in LSECs of both NAFLD animal models and patients [7,133]. The loss of autophagy in hepatic endothelial cells of Atg5lox/lox mice fed an HFD led to an induced expression of pro-inflammatory chemokines CCL2, CCL5, Cd68, VCAM-1, and cleaved caspase-3 and a significant decrease in the porosity and number of fenestrae of LSECs after mild acute liver injury [134,135]. A study by Hernández-Gea et al., revealed that pharmacological (using chloroquine (CQ)) and genetic (shRNA Atg7) downregulations of autophagy in LSECs isolated from untreated Sprague Dawley rats increased oxidative stress, as indicated by increased O_2_ levels [135]. The selective loss of endothelial autophagy in Atg7 endo mice after CCL4-induced liver injury resulted in endothelial cell dysfunction and reduced intrahepatic NO levels. In particular, autophagy seems to preserve the LSEC phenotype, at least in part, by regulating NO, handling oxidative stress, and maintaining cellular homeostasis [135]. On the other hand, a disruption of autophagy resulted in an aberrant antioxidant response through the reduced expression of antioxidant genes, including Nqo1, Hmox1, Gstm, Gclc, Gclm, and Srxn1, and NO production, along with hepatic ROS accumulation [2,135]. The other outstanding feature of LSECs is their endocytosis ability, which is the process of cleaning soluble macromolecules and smaller particles [136,137]. Both macrophages and pinocytes have been charged with the clearance of the circulating waste in the body. Therefore, LSECs express various endocytosis receptors, such as the mannose receptor (MR), scavenger receptor (SR), and Fc gamma-receptor IIb2 (FcRIIb2). SRs, including SR-A, SR-B, and SR-H (stabilin-1 and stabilin-2), are expressed in normal LSECs, and they are responsible for removing modified proteins and lipoproteins, including oxidized low-density lipoproteins (ox-LDL), as well as for extracellular matrix macromolecule uptake and degradation [136,138]. The expression of hepatic MR is regulated by inflammatory stimuli and cytokines. MR mediates glycoprotein uptake and lysosomal enzyme recruitment. It has been shown that the MR expression in LSECs is reduced by IL-10, while IL-1 increases its expression and activity, indicating the possible role of MR in the regulation of the LSEC inflammatory response. Finally, LSECs exert their unique ability to clear soluble IgG antigens and small soluble immune complexes from the circulation through the FcγRIIb2 receptor [136,138]. Of note, negative correlations between LSEC FcγRIIb expression and serum levels of blood triglyceride (TG), total cholesterol (TC), high-density lipoprotein cholesterol (HDL), and type 4 collagen (marker of fibrosis) were observed in human NAFLD biopsies [139].

### 3.4. LSECs in NAFLD-Related HCC

LSECs have been also implicated in the progression of NAFLD to HCC, although the available data are scarce. In 2009, Milner et al., reported that the adipokine fatty acid-binding protein 4 (FABP4) was elevated in NAFLD patients without HCC versus healthy controls, distinguishing steatohepatitis from simple steatosis and predicting liver inflammation and fibrosis [140]. Recently, it was demonstrated that FABP4 was overexpressed in human HCC samples from patients with metabolic syndrome, and this expression was mainly detected in peritumoral endothelial cells [141]. Interestingly, though FABP4 is not expressed by LSECs under basal conditions, the exposure of these cells to conditions mimicking NAFLD (high concentrations of glucose, insulin, and VEGFA) led to a significant release of FABP4 protein, and FABP4 increased the cell viability and proliferation of hepatocytes, leading to the conclusion that FABP4 exerts pro-oncogenic effects [141].

### 3.5. Novel Markers of LSEC Dysfunction

While various markers such as CD32b, CLEC4G, LYVE1, and STAB2 have emerged for the detection of LSECs in healthy livers, electron microscopy remains the gold standard for the identification of damaged LSECs, as it can detect the loss of basement membrane and fenestrae [142].

Recently, single-cell transcriptomic (scRNAseq) analyses in both healthy and diseased human and mouse livers have identified heterogeneity within the LSEC population and several biomarkers to assess disease development [142,143].

A transcriptome analysis of more than 100,000 single human cells revealed seven distinct endothelial subpopulations that inhabit in the fibrotic niche. These endothelial cells express both ACKR1^+^ and PLVAP^+^, which are restricted to cirrhotic liver tissue and induce the transmigration of leucocytes. Importantly, ACKR1 knockdown attenuated leucocyte recruitment by cirrhotic endothelial cells. A metagene signature analysis revealed that the expression of profibrogenic genes such as *PDGFD*, *PDGFB*, *LOX*, and *LOXL2* in the scar-associated endothelial cells was associated with extracellular matrix rearrangement and increased hepatic fibrillar collagens [144]. While the expression of LSEC-specific scavenger receptors, including *STAB2*, *CLEC4G*, *CD209*, *MRC1*, and *CD32B*, as well as receptors involved in VEGF-induced angiogenesis signaling, such as KDR and NRP1, have been defined as a signature of healthy LSECs [142], a study by Verhulst and colleagues showed that *STAB2* and *CLEC4G* are reduced during chronic liver diseases [142]. Transcriptomics revealed that the interaction between LSECs and chemokines was disrupted in mice deficient in STAB1 (Stab1KO) and STAB2 (Stab1KO) due to the reduced expression of adhesion molecules and other molecules involved in cytokine-cytokine receptor interaction [145]. Strong expression of both TIMP1 and TIMP2 in LSECs was observed in a single-cell analysis of human livers of both chronic and acute liver injury [142]. On the contrary, a study by Xiong et al., found no significant changes in the expression of both *TIMP1* and *TIMP2*, while they detected abundant expression of *Fcgr2b* and *Gpr182* by scRNAseq analysis [146]. The expression of endothelial cellular markers of lipid accumulation, *Cxcl9* and *BODIPY*, was strongly elevated in LSECs isolated from a trans-fat-containing amylin liver NASH (AMLN-diet)-induced NASH mouse model. A microarray dataset analysis of published data containing samples from 24 healthy, 20 NAFLD, and 19 NASH patients (GEO: GSE89632 [147]) by Xiong et al., showed that, similar to the NASH mouse, during human NASH pathogenesis there is a hepatic transcript abundance of *CXCL9* and *FABP4*, while the expression of *BMP2*, *NRP1*, and *VEGFA* is reduced in the hepatic tissues of patients with NAFLD and NASH [146].

Apart from the aforementioned markers of LSEC function/dysfunction, the expression of FABP4, fatty acid-binding protein 5 (FAPB5), von Willebrand factor (VWF), von Willebrand factor A domain-containing 1 (VWA1), and CD31 was found to be upregulated in LSECs during liver disease [49,142,148]. A single-cell RNAseq analysis by Verhulst and colleagues revealed the upregulation of FABP4/5 and VWF/a1 as a signature of damaged human LSECs [142]. The expression of FABP4 was found to be elevated during liver fibrosis. FABP4 promotes LSEC capillarization and therefore plays a crucial role during the onset and progression of liver fibrosis in mice [149]. The expression of VWF was not detected in healthy livers, while it was increased in LSEC fibrotic livers obtained from CCL4-treated mice and rats as well as NASH rats with or without cirrhosis [150,151].

## 4. Endothelial Cells as Therapeutic Targets for the Treatment of NAFLD

The complex and multifactorial pathogenesis of NAFLD makes the invention of a “wonder drug” not an easy task.

Currently, lifestyle modification and the management of its associated comorbidities remain the cornerstones of the treatment of NAFLD. As an approved pharmacological treatment for NAFLD is still missing, the identification of promising targets and the development of effective therapies is an urgent need [4].

Given their role in NAFLD pathogenesis and its related comorbidities and due to their specific biological characterization, endothelial cells could represent a “golden target” for the development of new treatment strategies [2]. Specifically, LSECs, as hepatic scavenger endothelial cells with an endocytosis capacity that enables them to capture soluble macromolecules and small particles through their numerous receptors, emerge as a suitable target for the development of new therapeutic approaches [7].

In this vein, basic and translational research using NAFLD animal models have targeted endothelial cells, providing promising results.

### Targeting LSECs in Experimental Studies

Statins are a class of 3-hydroxy-3-methylglutaryl CoA (HMG-CoA) reductase inhibitors used for the treatment of dyslipidemia and CVD due to their ability to reduce cholesterol synthesis. Recently, there is accumulating evidence indicating that statins have hepatoprotective effects through the alleviation of hepatic steatosis, NASH activity, and cirrhosis [152,153]. Statins exert beneficial effects on the liver through the improvement of endothelial impairment, increasing eNOS activity, the inhibition of Ras homolog family member A/Rho-associated coiled-coil forming kinases (RhoA/Rho-kinase), and the prevention of LSEC capillarization [152].

Pereira et al., demonstrated that simvastatin (SV) has a protective effect against hepatic and adipose tissue microcirculatory dysfunction in HFHC-fed mice. The improvement of microcirculatory disturbances by SV was due to decreased oxidative and ALE (advanced lipoxidation end products)-RAGE (receptor of advanced glycation end products) stress [154]. The same research group reported that SV treatment promotes blood flow recovery in microcirculation by the amelioration of sinusoid narrowing through decreased hepatic lipid accumulation and steatosis [58,154].

Since HSC activation also plays a crucial role in collagen deposition as well as in the regulation of vascular tone, the SV hepatoprotective effect against HSC activation may significantly reverse fatty liver progression [45]. SV treatment ameliorates fibrosis, as judged by positive α-SMA staining along with decreased collagen I mRNA levels in the liver of HFHC-fed mice [154]. In line with this study, Gracia-Sancho et al., demonstrated that SV treatment in diet-induced cirrhosis male Wistar rats reduces HSC activation and thereby significantly improves hepatic endothelium dysfunction and liver fibrosis, most likely through increased hepatic KLF2 expression [68]. In particular, SV-induced KLF2 expression, in both human and rat HSCs, was followed by HSC deactivation, a decrease in α-SMA and procollagen I expression, a restoration of sinusoidal cell capillarization, reduced oxidative stress, and an alleviation of endothelium dysfunction [68]. It should be mentioned that KLF2 is a known master regulator of cell phenotype that is responsible for the regulation of approximately 40% of the endothelium genome and exerts its beneficial effects on endothelial protection mainly through the induction of eNOS activity [65,68,155,156].

Data regarding the role of statins in experimental models of cirrhosis point towards a beneficial effect mediated via eNOS/NO signaling. In cirrhotic rat livers, SV treatment reduced oxidative stress and improved endothelial dysfunction and consequent liver damage through enhanced KLF2-dependent vasoprotective mechanisms [157]. Chronic treatment with atorvastatin lowered PP by decreasing intrahepatic resistance via the activation of eNOS/NO signaling in different experimental models of cirrhosis. Specifically, the treatment of cirrhotic rats with atorvastatin (15 mg/kg per day for 7 days) reduced PP without affecting mean arterial pressure in in situ perfused livers. Atorvastatin exerted these beneficial effects through the inhibition of hepatic RhoA/Rho-kinase signaling along with the activation of the NO/PKG pathway, which both decreased the intrahepatic resistance and PP [158]. Of note, the inhibition of the RhoA/Rho-kinase pathway in endothelial cells leads to elevated eNOS activation and expression through the phosphorylation of eNOS Ser1177 by Akt [158,159,160]. Treatment with NCX 6560 (a NO-releasing atorvastatin) showed a more profound beneficial intrahepatic effect through the induction of the p-eNOS/eNOS ratio compared to male Wistar rats under conventional atorvastatin treatment [161].

It was shown that SV treatment reduced both iNOS and collagen I expression and increased eNOS production in the diet-induced NAFLD mouse model and improved hepatic microcirculation impairment [162]. The iNOs and NO decreased by SV also account for the reduced HSC activation in the male Wistar rats with NASH-related hepatic fibrosis [162,163]. Of note, while the NO produced by eNOS exerts a hepatoprotective effect through the inhibition of the inflammatory activation of KCs, iNOS-derived NO has been shown to promote NAFLD [71]. Interestingly, SV increased eNOS activity and NO bioavailability, leading to reduced iNOS in rats after hepatic ischemia-reperfusion [162].

Bravo and co-workers demonstrated that atorvastatin and ambrisentan (a selective endothelin receptor-A antagonist) combination therapy normalized liver hemodynamics, reversed NASH histological features by 75%, and improved portal pressure in diet-induced NASH rats for 2 weeks. The authors reported that atorvastatin improved microvascular endothelial function, leading to a reversed sinusoidal contractile phenotype and reduced PP through the alleviation of insulin resistance and increased Akt phosphorylation and eNOS activity [69].

On the other hand, ambrisentan prevented HSC activation and the contractile response by blocking the ET-1 response [69]. Specifically, ambrisentan treatment blocked endothelin A (ETA) receptors, thereby increasing the ET-1 available to bind to the ETB receptors located in LSECs. It was previously shown that the activation of ETB receptors by ET-1 induces eNOS phosphorylation in LSECs, leading to vasodilatation [164]. Therefore, ambrisentan might indirectly contribute to the improvement of sinusoidal microvascular function [69]. Another study by the same research group showed that 2 weeks of treatment with SV reduced PP and induced vasoprotective effects in sorted hepatic cells isolated from Sprague Dawley rats under an HFGFD diet through the maintenance of HSCs in a quiescent state and the restoration of LSECs [152].

Open chromatin landscape profiling and a transcriptome analysis of whole liver and isolated LSECs derived from a diet-induced NASH mouse model showed increased VCAM-1 levels as a result of the observed lipotoxicity [165]. Therefore, targeting the VCAM-1 signaling pathway could lead to the alleviation of NAFLD. Indeed, the incubation of both mouse and human primary LSECs with palmitate acid (PA) resulted in the upregulation of VCAM-1 through the activation of the MAPK signaling pathway, as indicated by elevated MAP2K 3/6 (MMK3/6) and MAPK p38 phosphorylation. Interestingly, this effect was abolished after the pharmacological inhibition of either mitogen-activated protein 3 kinase (MAP3K) mixed lineage kinase 3 (MLK3) by URMC-099 or p38 by SB203580 [165].

In the same direction, a VCAM-1Ab treatment of FFC-fed mice showed a relative attenuation of inflammation, as indicated by reduced TNF-α, IL-1β, CD36, and CCR2 mRNA levels, as well as all injurious features of NASH compared to the IgG-treated mice. Moreover, pro-inflammatory macrophage (MoMF) populations were significantly reduced after VCAM-1Ab treatment [165]. Of note, an increased MoMF population contributes to the induction of inflammation and liver fibrosis during NASH. Interestingly, the hepatoprotective effect of VCAM-1 neutralization was also confirmed using mice deficient in hepatic endothelial VCAM-1 expression [165,166]. Another study from the same group demonstrated that the pre-treatment of LSECs with a neutralizing antibody against VCAM-1 reduces the adhesion of lipotoxic hepatocyte-derived extracellular vesicle (LPC-EV)-stimulated monocytes to LSECs. Of note, the hepatocyte-derived EV gradient is elevated in the liver microenvironment, mostly in the sinusoidal space, and is responsible for the activation of the liver homing signal in response to a lipotoxicity-induced injury. LPC-EVs are enriched with active integrin b1 (ITGb1) through which they interact with its ligand, VCAM-1, on the surface of LSECs. The treatment of diet-induced NASH mice (C57BL/6J under FFC treatment) with an anti-ITGb1 neutralizing antibody (ITGb1Ab) or ITGB1 knockdown reduced hepatic monocyte infiltration and MoMFs activation, both in vivo and in vitro, leading to an amelioration of hepatic inflammation, as suggested by reduced inflammatory markers such as TNF-a, CCR-2, and CD36. The macrophage polarization was also altered towards an M2 (anti-inflammatory) profile after the treatment of mice with ITGb1Ab [167]. Thus, ITGb1Ab exerts hepatoprotective effects by the attenuation of liver injury, inflammation, and fibrosis through the blocking of signaling molecules responsible for monocyte adhesion to LSECs [165,167]. Strengthening the potential role of VCAM-1 as a therapeutic target, an animal study showed that the administration of succinobucol, a VCAM-1 pharmacological inhibitor (AGI-1067), for the treatment of the advanced stages of NASH in mice resulted in improvements in insulin resistance, inflammation, and ultimately liver injury. Therefore, VCAM-1 blockade can provide a potential therapy for NASH through various mechanisms, amongst them the reduction in pro-inflammatory monocyte infiltration into the liver. It is noteworthy to mention that succinobucol has been employed in clinical trials for the treatment of atherosclerosis and type 2 diabetes patients with CVD. However, its effect on NAFLD and liver diseases remains mostly unknown, as patients with moderate to severe hepatic dysfunction were excluded from these clinical studies [168,169].

PPAR-α is known as a general modulator of the inflammatory response and a negative regulator of molecules such as ET-1, VCAM-1, and inflammatory cytokines, i.e., IL-6, in the endothelium [170,171]. Specifically, PPAR-α exerts anti-inflammatory effects by inhibiting the transcriptional activities of pro-inflammatory transcription factors, including nuclear factor kappa B (NF-kB), activator protein 1 (AP-1), and signal transducer and activator of transcription (STAT) [172]. The treatment of HFD-fed Foz/Foz mice with Wy-14643, a potent PPAR-a agonist, exerts a hepatoprotective effect by reducing the expression of inflammatory markers and adhesion molecules and reducing hepatic lipid accumulation in NASH mice subjected to 60 min of ischemia and 15 min of reperfusion [62]. The treatment of NASH mice with Wy-14,643 reduced the expression of VCAM-1, IL-1α, TNF-α, and IL-12, most likely through the rapid activation of the NF-kB and p38 pathways [62]. Interestingly, Wy-14,643 treatment had no significant effect on ICAM-1 expression in the livers of either steatotic or NASH mice. On the contrary, the activation of the NF-kB pathway in MCD-diet-induced steatohepatitis in TNF−/− and TNFR-1−/− mice [173] as well as in the MCD-fed wild-type mice resulted in the upregulation of ICAM-1 expression [174].

High-mobility group box 1 (HMGB1) is a known inflammatory cytokine that interacts with various receptors, including TLR2, TLR4, TLR9, and RAGE, leading to the stimulation of hepatic oxidative stress, the inflammatory response, and finally endothelial dysfunction [175,176]. The expression of HMBG-1 has already been reported to be closely associated with the development of liver fibrosis. Hepatic HMGB1 protein levels were induced in the mouse models of NASH. An HMGB1-neutralizing antibody prevented liver inflammation and fibrosis, while an injection of recombinant HMGB1 promoted liver fibrosis in mice treated with CCl4 for one month [6].

Hepatic angiogenesis promotes chronic inflammation in NASH, while the inhibition of angiogenesis has been shown to improve hepatic inflammation. Hepatic angiogenesis is regulated by both vascular endothelial growth factor (VEGF/VEGFR) and angiopoietin/tyrosine kinase with immunoglobulin-like and EGF-like domains 2 (Ang/Tie2) pathways [177,178]. Under physiological conditions, VEGF is secreted by hepatocytes and quiescent HSCs as a paracrine signal, and it is crucial for the maintenance of LSEC fenestration. VEGF overexpression is stimulated under hypoxic conditions in NASH due to the increased oxygen consumption used for lipid metabolism, leading to induced mechanical pressure on the sinusoids [58,179]. The administration of specific antibodies against VEGF receptor 2 (VEGFR2) in the MCD-induced NASH mouse model reduced liver inflammation and improved hepatic microcirculation [180]. The administration of the anti-VEGFR antibodies known as brivanib (3 mg/kg/day) and sorafenib (5 mg/kg/day) for two weeks to the diet-induced NASH-cirrhotic rat model (fed HF/MC diet for 12 weeks) prevented hepatic blood flow reduction and reduced the expression of inflammatory cytokines such as TNF-α, IL-1β, IL-6, and IL-17 [181]. CD31 overexpression was observed in LSECs isolated from fibrotic liver, while LSEC CD31 surface expression was prevented when they were co-cultured with HSCs or hepatocytes incubated with anti-VEGF antibodies [90].

Lefere and colleagues showed that the inhibition of the Ang-2/Tie2 interaction by peptibody L1–10 alleviates ballooning and fibrosis as well as hepatic inflammation, angiogenesis, and microvascular architecture distortion in MCD-diet-induced NASH and streptozotocin-western-diet-induced NASH mouse models [178]. The hepatoprotective effect of L1–10 therapy seems to be, at least in part, through its effects on LSECs since L1–10 reduced the expression of VCAM-1, ICAM-1, and MCP1 in LSECs isolated from mice fed a methionine- and choline-deficient diet. Interestingly, blocking the Ang-2 signaling in the streptozotocin-western diet NASH mouse model resulted in reduced VCAM-1 expression, reversed NASH, and ameliorated HCC progression. Additionally, an in vitro treatment of LPS-stimulated LSECs with L1–10 led to decreased expression of inflammatory markers [177,178].

ITGα4β7 regulates the binding of lymphocytes to its endothelial ligands, MAdCAM1 and VCAM-1 [177]. The treatment of the diet-induced NASH mouse model with an ITGα4β7 antibody reduced liver inflammation, fibrosis, and metabolic dysfunction, as the ITGα4β7 antibody reduced hepatic CD4^+^ T-cell homing [182]. During CCL4-induced liver fibrosis, naïve CD8+ T-cell populations were induced in WT mice, leading to increased fibrosis and HSC activation. The effect was diminished after subtotal irradiation with a single total-body dose of 700 cGy [183].

A study by Mendt et al., revealed that LSEC conditional medium provoked the migration of BM progenitor lineage-negative (BM/Lin^−^) cells isolated from WT mice. This effect was abolished after the incubation of cells with a CXCR4 inhibitor known as AMD3100 [184]. Moreover, AMD3100 treatment reduced the CD4+ T cell number and abolished the chemotactic effect of CXCL12 on CD4+ T cells in a NASH mouse model, indicating that CXCR4 can represent a potential therapeutic target for NASH treatment [110]. On the contrary, a gavage treatment of fibrotic mice with AMD070, another CXCR4 inhibitor, exerted a negligible effect on the liver fibrosis markers [185].

The treatment of primary LSECs isolated from mice with PA induced the expression of NOX-1, while this induction was repressed after the incubation of cells with TAK-242 (TLR4-inhibitor) [80]. Of note, NOX-1 induces liver fibrosis through the regulation of hepatocyte proliferation and ROS generation [80,186].

Currently, nanoparticle (NP) drug delivery methods have attracted tremendous attention as a therapeutic approach for the treatment of NAFLD. NPs represent an opportunity to achieve sophisticated targeting therapy due to their size and surface characteristics and their ability to protect drug degradation and control the drug cellular uptake at the desired tissues, such as the liver [187]. Due to the lack of basal lamina and the size of liver sinusoidal fenestrae (50–200 nm), LSECs provide a mesh-like structure, contributing to the entrapment of NPs in the liver. This structure facilitates the accumulation of high concentrations of NPs in the Disse space and their distribution to other liver cells [188]. 

After systemic administration, NPs with a size greater than 6 nm accumulate in the liver. Therefore, enterohepatic microcirculation plays a crucial role in delivering orally administered NPs to the liver [189]. Recently, an apolipoprotein B (ApoB) sequence has been used to decorate nanoparticles, given that ApoB is a ligand for both scavenger receptors, stabilin-1 and 2, expressed by LSECs [138].

Hyaluronic acid (HA) micelles targeting HA receptors can be used for targeting both LSECs and HSCs. HA micelle particles carrying losartan were shown to be an effective NP delivery system that ameliorates advanced liver fibrosis in a C3H/HeN mouse model, as demonstrated by reduced ALT and AST serum and decreased hepatic alpha smooth muscle actin (α-SMA) expression [190]. Moreover, lipid NPs carrying procollagen α I(I) siRNA remarkably reduced the total hepatic collagen content, leading to an alleviation of NASH progression and accelerating the regression of hepatic fibrosis in CCL4-induced NASH in Balb/c mice [191]. In another attempt, CXCR4-targeted lipid-based NPs carrying VEGF siRNA were used for the treatment of liver fibrosis and hepatic cellular carcinoma; a combination therapy using both AMD-NPs (CXCR4 inhibitor) and VEGF siRNA abolished the infiltration of tumor-associated macrophages and reduced HCC tumor growth [192].

Apart from the above-mentioned methods, the use of quantum dot (QD) nanoparticles (1–20 nm) for the treatment of NAFLD has also emerged as an area of great interest [193]. An intravenous injection of mercaptosuccinic acid (MSA)-capped cadmium telluride/cadmium sulfide (CdTe/CdS/QDs) showed that negatively charged QDs were selectively taken up by sinusoidal cells (KCs and LSECs) in rat liver, indicating that CdTe/CdS/QDs can be used for drug delivery to the LSECs [194]. Furthermore, Zn-labelled CdSelenide/CdS/ZnS QDs were also found in the KCs and LSECs 2h after an intravenous injection of polymer-coated Qdots, indicating their efficiency in drug delivery to the sinusoidal cells [195]. In addition, via the usage of nanoparticles carrying honokiol or adenovirus with the endothelial-cell-specific arginylglycylaspartic acid-roundabout guidance receptor 4 (RGD-ROBO4), ERK1/2 activation was promoted, leading to induced liver regeneration and reduced hepatic fibrosis due to high degree of LSEC specificity [196].

The main studies targeting endothelial dysfunction in animal models of NAFLD/NASH/cirrhosis are summarized in Table 1.

## 5. Vascular Endothelial Dysfunction and NAFLD: Human Studies

Although NAFLD can progress to NASH, cirrhosis, and finally HCC, CVD is the main cause of mortality [197,198,199]. NAFLD has been associated with elevated risk for coronary heart disease and atherosclerosis progression [200,201,202]. Most importantly, NAFLD occurrence leads to a higher risk of vascular endothelial dysfunction and atherosclerosis, independent of the metabolic syndrome and its other manifestations [12].

NAFLD could be implicated in the increase in atherosclerotic and CVD risks since the disease can lead to vascular endothelial dysfunction through the local overexpression of inflammatory mediators and regulators of arterial blood pressure and atherogenic dyslipidemia (an elevation in triglyceride-rich lipoproteins, such as very-low-density lipoproteins (VLDL), increased small dense low-density lipoproteins (LDL), and decreased concentrations of high-density lipoproteins (HDL)) [198,203]. Additionally, the insulin resistance observed in NAFLD can lead to vascular endothelial dysfunction through various mechanisms, including an imbalance in NO production, that can lead to decreased blood flow, which in turn worsens insulin resistance, creating a vicious cycle [198,204]. The perturbation of oxidative stress, a phenomenon widely observed in NAFLD, has been shown to affect metabolic inflammation by causing vascular endothelial dysfunction, thus increasing the risk for NAFLD patients to develop CVD [205]. A variety of clinical studies have tried to assess the presence of endothelial dysfunction in patients with NAFLD (Table 2). The most commonly used markers in these studies are brachial flow-mediated vasodilation (FMD), carotid artery intima-media thickness (CIMT), and the serum levels of asymmetric dimethylarginine (ADMA), one of the results of methylated protein degradation that displays an inhibitory effect on eNOS, thus leading to reduced levels of NO and consequently endothelial dysfunction [206]. NAFLD diagnosis was performed via ultrasound criteria and/or liver biopsy, while in a few studies magnetic resonance spectroscopy or a controlled attenuation parameter were used to determine liver steatosis [12,207,208,209,210,211,212,213,214,215,216,217,218,219,220,221,222,223,224,225,226,227,228,229,230,231,232]. However, only a few studies have investigated the progress of endothelial dysfunction with regard to the presence and severity of NASH [12,199,208,212,221,226,229].

In the majority of these studies, patients with NAFLD exhibit impaired FMD, even though the brachial artery diameter at baseline shows no statistical differences between patients with NAFLD and controls. On the contrary, the results regarding CIMT are contradictory since NAFLD patient CIMT is found to be higher in some studies, while in others no statistically significant differences were found when adjusted for factors of metabolic syndrome, such as obesity and insulin resistance. As far as ADMA is concerned, discrepancies are also found, with some studies showing higher levels in patients with NAFLD, while in others no differences were found.

In the largest of the aforementioned studies, encompassing 350 patients with NAFLD and 1934 controls from the Framingham study cohort, lower FMD and peripheral artery tonometry (PAT) ratios and higher carotid-femoral pulse wave velocities (PWV) were found in NAFLD patients. However, only PAT was proven to be statistically significant after adjusting for mean arterial pressure, further complicating the presence and diagnosis of vascular endothelial dysfunction in patients with NAFLD [211]. Another cross-sectional study investigating the relationship between whole blood viscosity (WBV) and NAFLD in 1329 subjects revealed that the WBV at low shear stress was elevated in patients with NAFLD [233].

### Liver-Secreted Molecules Leading to Vascular Endothelial Dysfunction

The liver has been considered a significant source of pro-inflammatory factors, such as TNF-α, IL-6, interleukin-1β (IL-1β), CRP, fibrinogen, and fetuin A [198,205]. In fact, several inflammatory markers, such as C-reactive protein (CRP), IL-6, monocyte chemotactic protein 1 (MCP-1), and TNF-α, have been found to be elevated in patients with NAFLD and even more exacerbated in patients with NASH compared to healthy controls, independent of obesity and other confounding factors. However, data on specific soluble factors that are secreted from the liver and act on vascular endothelial cells, leading to vascular endothelial dysfunction, are scarce. In such an attempt, Baldini et al., tried to identify such molecules produced during hepatic steatosis that could lead to endothelial dysfunction without cell-to-cell contact [234]. In order to replicate the hepatic steatosis caused by a high-fat diet in vitro, FaO cells (rat hepatoma cell line that retains hepatocyte-specific markers) were treated with different combinations of oleate/palmitate (OP), TNF-α, and fructose with fatty acids and then the conditioned medium from the steatotic hepatic cells was used to treat human endothelial cell line from umbilical vein (HECV) cells [234]. The treatment with the medium from the steatotic hepatocyte culture induced an increase in fat accumulation and ROS and NO levels as well as an upregulation in the mRNA expression of endothelial PPAR-γ and VCAM-1 and the oxidative stress markers Zn/superoxide dismutase (SOD) and Sirtuin-1 (Sirt-1) [234]. Steatotic hepatocytes seem to release lipids and mainly triglycerides, which can trigger lipid accumulation in vascular endothelial cells, leading to fat-dependent dysfunction and oxidative stress, possibly through other soluble mediators as well, ultimately promoting pro-inflammatory and pro-atherogenic effects [234].

Another molecule that has emerged as a factor leading to endothelial dysfunction and subclinical atherosclerosis in the setting of NAFLD is circulating fetuin-A [199]. Fetuin-A is a glycoprotein produced in the liver, which is then secreted in high concentrations into circulation and leads to insulin resistance through inhibiting insulin receptor signal transduction [199]. Interestingly, in humans, elevated fetuin-A levels have been associated with metabolic syndrome and its manifestations, namely, obesity, insulin resistance, risk of type 2 diabetes development, and with NAFLD as well as the risk of myocardial infarction and ischemic stroke [199]. When NAFLD male patients were compared to healthy control subjects, fetuin-A and the endothelial dysfunction marker asymmetric dimethyl arginine (ADMA) were higher, while the endothelial dysfunction marker adiponectin was lower in the NAFLD group that also displayed increased carotid intima-media thickness (cIMT), an early marker of subclinical atherosclerosis [199]. A correlation analysis indicated that fetuin-A was positively correlated with triglycerides, insulin resistance, ADMA, and cIMT and negatively correlated with high-density lipoprotein cholesterol (HDL-C) and adiponectin levels, while a multiple linear regression analysis showed that fetuin-A was independently associated with ADMA and cIMT levels [199]. These data point towards a role of fetuin-A in the development of endothelial dysfunction and subclinical atherosclerosis in the context of NAFLD [199].

In another study by Kasumov et al., ceramide production was found to be implicated in the development of diet-induced NAFLD and associated atherosclerosis since the inhibition of this production protected from these complications [197]. In particular, low-density lipoprotein receptor deficient (LDLR−/−) mice, an established diet-induced model of both NAFLD and atherosclerosis, which are susceptible to diet-induced hepatic inflammation and fibrosis, which can lead to the progression of simple steatosis to NASH and the development of atherosclerosis, were fed a western-type diet and treated with a pharmacological inhibitor of sphingolipid biosynthesis [197]. While the administration of the western-type diet caused hepatic oxidative stress, inflammation, and apoptosis, the pharmacological inhibition of ceramide biosynthesis improved the induced insulin resistance, ameliorated atherosclerosis, hepatic steatosis, and fibrosis, and reduced apoptosis without any effect on oxidative stress [197]. Liver-derived ceramides with triglycerides and cholesterol have been proposed to be packaged with VLDL and released into circulation [197]. Indeed, ceramides are carried in VLDL and LDL particles, as shown by lipoprotein fraction analysis [235]. Ceramide-loaded LDL has been found to activate NF-kΒ signaling and through this to increase the expression and secretion of the inflammatory molecules TNF-α and IL-6 in macrophages [236]. Based on these findings, ceramides have been suggested to induce cytokine production and release into the circulation, which can then target endothelial cells and promote atherosclerosis via the inflammatory process [197]. Furthermore, other suggested mechanisms regarding the role of ceramides in promoting vascular endothelial dysfunction include the enhanced susceptibility to aggregation and retention in the arterial wall that can be caused by the increased ceramide content in oxidized remnant lipoproteins and the reduced NO bioavailability caused by the circulating ceramides [197].

Selenoprotein P (SeP) is a secretory protein primary produced in the liver that causes insulin resistance in humans [237], and its serum levels have been found to be significantly elevated in NAFLD patients compared to healthy controls [209]. Interestingly, consistent with the association between elevated circulating SeP levels and atherosclerosis [238], higher serum SeP levels in NAFLD patients have been positively correlated with body mass index (BMI), fasting glucose, LDL cholesterol, and insulin resistance and negatively correlated with flow-mediated dilation (FMD), which has been observed to be lower in NAFLD patients compared to controls [209]. These associations indicate that SeP may be related to insulin resistance and its metabolic consequences, rendering SeP as a possible link between NAFLD and the pathogenesis of the associated endothelial dysfunction [209].

## 6. Conclusions

Animal and human studies point towards a strong interrelation of NAFLD and atherosclerosis, with the existence of one increasing the risk of the other. Common pathogenetic mechanisms such as low-grade inflammation, insulin resistance, oxidative stress, and hyperlipidemia seem to be implicated in the progression of both NAFLD and atherosclerosis, with endothelial cells (vascular and LSECs) emerging as key players. Although molecules secreted by liver have been recognized to lead to vascular endothelial dysfunction, whether there is broad distant intercellular communication between vascular and LSECs that drives this interplay in both directions warrants intensive investigation.

Endothelial cells could represent a “golden target” for the development of new treatment strategies for both conditions. Specifically, LSECs, as hepatic endothelial cells with specific locations and distinct properties that enable them to capture macromolecules and small particles through their numerous receptors, emerge as a suitable target for the development of new therapeutic approaches for NAFLD. Experimental studies in this direction need to be effectively translated to clinical practice, thereby allowing researchers to investigate a possible causative role of NAFLD in the atherosclerosis process and/or vice versa.

## Figures and Tables

**Figure 1 cells-11-02511-f001:**
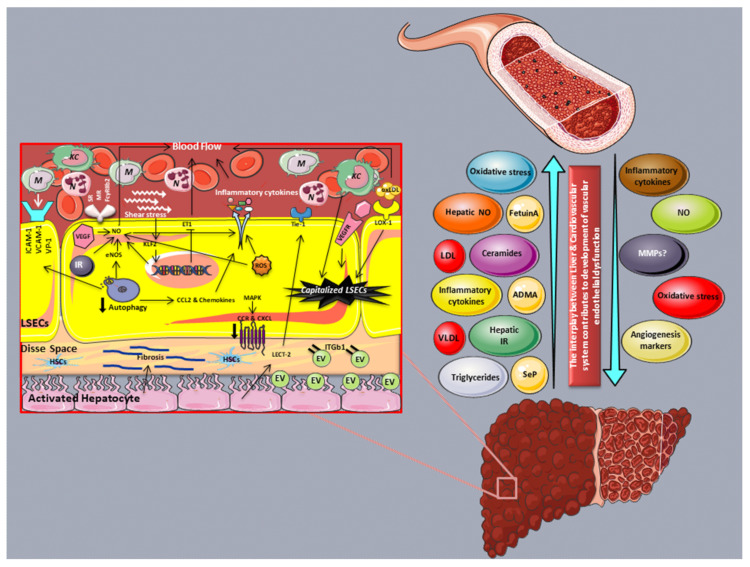
The role of endothelial cells in NAFLD pathogenesis and the interplay between CVD and NAFLD. LSECs are located at the interface between the blood stream and the liver parenchyma. LSECs regulate blood flow in response to shear stress, mainly through increased NO synthesis and bioavailability as well as through ET-1 reduction, which are both mediated by KLF2. LSECs also regulate the activation of KCs and HSCs during NASH progression. The expression of SRs, MR, and FcγRIIb2 endows LSECs with high endocytic capacity; of note, the reduced endocytic capacity of LSECs precedes fibrosis in NAFLD. The increased expression of adhesion molecules during the early stage of NAFLD enhances the recruitment of monocytes to the inflamed endothelium, leading to the activation of an inflammatory response in NASH. Impaired autophagy has been associated with the development of steatosis and fibrosis through—among others—the upregulation of adhesion molecules and pro-inflammatory mediators during the progression of the disease. Moreover, hepatocyte-derived EVs contribute to the formation of inflammatory foci by the recruitment of macrophages into the hepatic sinusoids. Both HSC- and LSEC-derived EVs play crucial roles in the maintenance of the balance between extracellular matrix production and degradation and the consequent progression towards the regeneration of hepatic cells or fibrosis. The anti-inflammatory features of LSECs observed during the early stage of NAFLD development are attributable—among others—to decreased CCL and CXCL expression through MAPK signaling activation and increased secretion of IL-10 by Th1 cells. NAFLD is strongly related to vascular endothelial dysfunction and consequence atherosclerosis. The overexpression of inflammatory mediators, elevated insulin resistance, and oxidative stress are key players in this interrelation. Increased levels of inflammatory molecules such as circulating fetuin-A, ADMA, cRP, and SeP have been associated with an elevated risk of CVDs in NAFLD patients and vice versa: the low-grade inflammatory milieu of atherosclerosis could promote the progression of NAFLD.

**Figure 2 cells-11-02511-f002:**
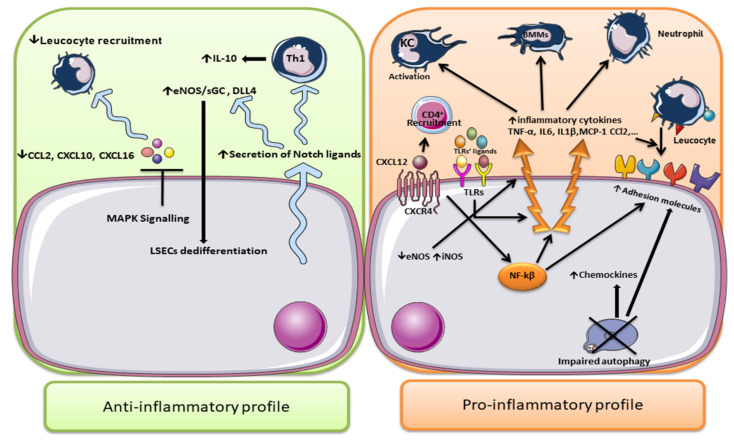
The LSECs’ anti-inflammatory and pro-inflammatory profiles during the progression of NAFLDAt the early stage of NAFLD, LSECs display an anti-inflammatory function characterized by a reduced expression of chemokines such as CCL2, CXCL10, and CXCL16 through MAPK signaling and an induced expression of IL-10 by Th1 cells through the activation of Notch signaling. The activation of Notch signaling manifests anti-inflammatory effects through the induction of eNOS/sGC levels.During the progression of NAFLD from simple steatosis to NASH and cirrhosis, LSECs exhibit a pro-inflammatory phenotype mediated mostly through the activation of the NF-kB pathway. NF-kB regulates the expression of adhesion molecules (VCAM-1, ICAM-1, E-selectin, and VAP-1) as well as the secretion of pro-inflammatory cytokines (TNF-α, IL-1, and IL-6). The secretion of inflammatory mediators is also regulated by TLRs and NO bioavailability. Elevated expression of adhesion molecules leads to induced leukocyte recruitment and their translocation into the hepatic parenchyma. On the other hand, increased expression of inflammatory mediators along with LSEC dysfunction stimulates the activation of KCs and leukocyte chemoattraction. The impaired LSEC autophagy observed during the progression of NAFLD also leads to the upregulation of adhesion molecule and chemokine expression, enhancing the inflammatory response. Reduced eNOS and increased iNOS contribute to the development of inflammation, the activation of KCs, and the recruitment of bone-marrow-derived macrophages. Abbreviations: ICAM-1: Intercellular adhesion molecule-1; IL-1: Interleukin 1; IL-6: Interleukin 6; LSECs: Liver sinusoidal endothelial cells; MCP1: Monocyte chemoattractant protein-1; NF-kB: Nuclear factor kappa B; NO: nitric oxide; TNFa: Tumor necrosis factor alpha; VAP-1: Vascular adhesion protein1; VCAM-1: Vascular cell adhesion molecule-1; BMMs: bone-marrow-derived macrophages; TLR: Toll-like receptor; CXCL12: C-X-C motif chemokine ligand 12; CXCR4: C-X-C chemokine receptor type 4; KC: Kupffer cells. This image was derived from the free medical site http://smart.servier.com/ (accessed on 1 July 2022) by Servier, licensed under a Creative Commons Attribution 3.0 Unported license.

**Table 1 cells-11-02511-t001:** Studies targeting endothelial cells as a therapeutic strategy in animal models of NAFLD/NASH/HCC.

Ref.	Animal Model	Treatment	Markers of NAFLD	Markers of Endothelial Dysfunction	Outcome after Therapy
[62]	foz/foz mice under HFD/chew diet	PPAR-alpha agonist (Wy-14,643) (10 days)	ALT Hepatic necrosis area Inflammation (IL-1a, TNF-a, IL-12)	Adhesion molecules (VCAM-1) Microvascular narrowing	⬇ ALT ⬇ Adhesion molecules ⬇ Inflammation ⬇ Fibrosis ⬇ Microvascular narrowing
[69]	Male Sprague Dawley rats under HFGFD	Atorvastatin and/or Ambrisentan (2 weeks)	ALT Contractile response in HSCs NA Score	Portal hypertension eNOS	Combination therapy normalizes liver hemodynamics and reverses HSC procontractile and profibrogenic profile
[152]	Sprague Dawley rats under HFGFD for 8 weeks	Simvastatin or Atorvastatin (2 weeks)	NA Score Lipid droplet HSC activation	Portal pressure CD32b/CD11b ratio Contractile phenotype	Reverses NASH histology features ⬇ LSEC differentiation ⬇ LSEC capillarization
[153]	C57BL/6 mice received ip CCl4 injection 2/W for 4 weeks	Simvastatin by tail vein injection and simvastatin-free drug daily (5 days)	HSC activation	CD31 KLF2-NO signaling CXCL16	Restores the quiescence of activated HSCs Alleviates LSEC capillarization Induces NKT recruitment into HCC microenvironment through CXCL16
[154]	C57BL/6 mice under HFHC diet	Simvastatin For 5 weeks	ALT and AST Serum and hepatic TG and TC NA score Hepatic fibrosis	Microcirculatory dysfunction Microvascular blood flow Leukocyte recruitment	Restores the endothelium-dependent vasodilatory response and blood flow Restores antioxidant enzymatic activity ⬇ Lipid peroxidation
[162]	Wistar rats under HFD models	Simvastatin (4 weeks)	ALT and AST Fibrosis α-SMA and Collagen I TC and TG	iNOS eNOS	Improves NASH-related fibrosis by increasing ⬆ eNOS ⬇ iNOS
[158]	Male Sprague Dawley rats underwent BDL	Atorvastatin (7 days)	ALT and AST HSC activation	eNOS Portal pressure	⬆ eNOS ⬆ NO/PKG pathway activation ⬇ Portal pressure
[165]	C57BL6/J mice under FFC and chew diets and endothelial-cell-specific Vcam1 knockout mice	VCAM-1-neutralizing Ab or MLK3 inhibitor	ALT NA score Fibrosis	Adhesion molecules such as VCAM-1 Inflammatory markers such as IL-1b, CCR2, TNF-a, and CD68	⬇ Inflammation ⬇ Improved NA score ⬇ Fibrosis
[174]	C57BL6/J mice under MCD treatment	Curcumin (NF-kappaB inhibitor) (up to 4 weeks)	ALT NA Score Lipoperoxides, Pro-inflammatory markers such as COX-2, MCP-1, and CINC	Adhesion molecules such as ICAM-1	⬇ Inflammation ⬇ Liver histological features
[6]	C57BL6/J mice under MCD treatment	Anti-HMGB1 antibody	ALT NA score Fibrosis	CD31	Prevents liver fibrosis
[180]	C57BL6/J mice and db/db mice under MCD	Anti-VEGFR2 antibody (2 weeks)	ALT and AST HSC activation Fibrosis	Inflammation, CD105	Prevents liver fibrosis ⬇ Inflammation ⬇ Disrupted vascular architecture
[178]	C57BL6/J mice under MCD, C57BL6/J mice injected with strepto-zotocin (STZ), and C57BL6/J mice under WD	Ang-2/Tie2 interaction inhibiting peptibody L1–10	ALT and AST Inflammation, MCP-1, CXCL2, and CCR2 Fibrosis	Adhesion molecules ICAM-1 and VCAM-1 CD34 Ang-2	Reverses NASH ⬇ HCC progression ⬇ Inflammation ⬇ Liver histological features ⬇ Fibrosis ⬇ iNOS
[184]	LSECs isolated from C57BL6/J mice	AMD3100 (CXCR4 antagonist)	HSC migration	CXCR-4 expression under treatment with inflammatory cytokines PECAM-1	⬇ Inflammation ⬇ CXCR4 ⬇ CXCl12 ⬇HSC migration
[196]	C57BL/6J mice under CCl4 and LSECs isolated from C57BL/6J mice under CCl4	NP carrying honokiol or Ad-CA-MEK1 Ad-DN Akt	HSC activation HSC senescence	LSEC profibrotic phenotype NO/ROS	promotes Erk1/2 activity ⬇ Liver fibrosis ⬆ Enhances liver regeneration

Abbreviations: ALT: alanine aminotransferase, AST: aspartate aminotransferase, TG: triglycerides, TC: total cholesterol, IL: interleukin, TNF-α: tumor necrosis factor, NFκΒ: nuclear factor kappa-light-chain-enhancer of activated B cells, MCP-1: monocyte chemoattractant protein-1, PPAR-α: peroxisome proliferator-activated receptor alpha, Cxcl: chemokine (C-X-C motif) ligand, MMP: matrix metalloproteinases, HF: high fat, HFGHD: high-fat glucose-fructose diet, MLK3: MAP3K mixed lineage kinase 3, MCD: methionine/choline-deficient diet, VCAM: vascular cell adhesion molecule, ICAM: intercellular adhesion molecule, VEGF: vascular endothelial growth factor, Ad: adenovirus, ⬇: Decrease, ⬆: Increase.

**Table 2 cells-11-02511-t002:** Main studies regarding endothelial dysfunction in patients with NAFLD.

Ref.	Study Population (nr)	Method of NAFLD Diagnosis	Method of Endothelial Dysfunction Diagnosis	Outcome
[229]	39 NASH vs. 13 NAFL vs. 28 healthy controls	U/S, LB	FMD	FMD levels lower in NASH than all and lower in NAFL than controls
[226]	15 NASH vs. 17 NAFLD vs. 16 healthy controls	U/S, LB	FMD	Lower FMD in NASH but not in NAFLD
[225]	20 NAFLD with arterial hypertension vs. 20 hypertensive controls	U/S	FBF	FBF significantly reduced in NAFLD patients
[212]	70 NAFLD vs. 70 healthy controls	LB	PTX-3, ADMA	Higher PTX-3 and ADMA serum levels in patients with NAFLD, higher PTX-3 but not ADMA in NASH
[224]	100 NAFLD vs. 45 healthy controls	Biochemical, radiological and/or histological criteria	ADMA, FMD	No statistically significant differences
[227]	32 NAFLD vs. 16 healthy controls	U/S	FMD	Lower % of FMD change
[220]	93 MS+/NAFLD+ vs. 78 MS+/NAFLD− vs. 101 MS−/NAFLD−	FLI	Vasodilating response of FBF to acetylcholine	Worse response to vasodilation in MS+/NAFLD+ and MS+/NAFLD− patients
[210]	51 NAFLD vs. 21 healthy controls	LB	ADMA, FMD, CIMT	Higher CIMT and lower FMD in patients with NAFLD, no difference in ADMA
[216]	19 NASH vs. 19 NAFL vs. 19 healthy controls	U/S, LB	FMD	FMD levels lower in NASH than all and lower in NAFL than controls
[211]	24 NASH vs. 23 borderline NASH vs. 20 NAFL	LB	ADMA, CIMT	Higher ADMA in NAFLD, no differences in CIMT when adjusted for metabolic parameters and insulin sensitivity
[230]	23 NAFLD vs. 28 healthy controls	LB	PWV, CIMT, FMD	Higher PWV and CIMT and lower FMD in NAFLD
[231]	14 obese children with NAFLD vs. 14 obese children	MR spectroscopy	FMD	No difference
[214]	24 nondiabetic NASH vs. 11 nondiabetic NAFL vs. 25 healthy controls	LB	ADMA	Higher in NAFLD, no difference when adjusted for insulin resistance
[215]	117 NAFLD vs. 44 healthy controls	U/S	FMD, CIMT	Lower FMD in NAFLD correlated with U/S staging, higher CIMT in NAFLD, not correlated with U/S staging
[221]	12 NASH vs. 12 NAFLD vs. 28 healthy controls	LB	APDV, AMDV, CFVR, CIMT	2 min after dipyridamole infusion, APDV, AMDV, and CFVR were lower in patients with NAFLD, no differences between NAFL and NASH patients
[228]	40 nondiabetic NAFLD vs. 40 healthy controls	U/S	FMD, CIMT	Higher CIMT and lower FMD in NAFLD patients
[199]	115 NAFLD (50 NASH, 35 borderline NASH, 30 NAFL) vs. 74 healthy controls	LB	Fetuin-A, ADMA, adiponectin, CIMT	Higher ADMA, CIMT, and fetuin-A and lower adiponectin serum levels in NAFLD patients. No difference when the findings were adjusted according to the BMI, glucose, lipids, and HOMA-IR index
[208]	50 NASH vs. 30 healthy controls	LB	CIMT, FMD	Lower FMD and higher CIMT in NASH patients
[222]	34 obese NAFLD vs. 20 obese controls	Proton MR spectroscopy	FMD	Lower in NAFLD
[218]	49 NAFLD + DM vs. 50 NAFLD vs. 52 healthy controls	U/S	CIMT	Higher median CIMT in NAFLD patients when compared with controls irrespective of DM when adjusted for confounders
[217]	350 NAFLD vs. 1934 controls with no cardiovascular disease	CAP	FMD, PAT ratio, PWV	Lower FMD and PAT ratio and higher PWV in NAFLD patients. Only PAT ratio statistically significant after adjusting for mean arterial pressure
[12]	39 young men with NASH vs. 22 young men with NAFL vs. 41 young healthy control men	LB	PWV, FMD, CIMT	Lower FMD and higher PWV in NAFLD vs. controls, no difference between NAFL and NASH; higher CIMT in NASH vs. NAFL and controls
[223]	176 NAFLD vs. 90 non-NAFLD controls	U/S	FMD	Lower FMD in NAFLD patients, especially in patients with grade 3 steatosis
[209]	93 NAFLD vs. 37 healthy controls	U/S, LB	FMD, CIMT	Lower FMD in NAFLD, no statistically significant difference in CIMT
[207]	42 NASH vs. 47 NAFL vs. 50 healthy controls	LB	FMD	Lower FMD in NAFLD, especially in NASH
[219]	126 NAFLD (58 with MS, 68 without MS) vs. 31 CHB	U/S	FMD, CIMT	Lower FMD in NAFLD independent of MS; higher CIMT in patients with MS
[232]	95 NAFLD vs. 90 obese controls	C/T	FMD, NMD	Lower FMD in NAFLD, no differences in NMD
[213]	25 NAFLD vs. 25 healthy controls	U/S	FMD	Lower FMD in patients with NAFLD

Abbreviations: ADMA: asymmetric dimethylarginine; AMDV: average mean diastolic velocity; APDV: average peak diastolic velocity; BMI: body-mass index; CAP: controlled attenuation parameter; CFVR: coronary flow velocity reserve; CHB: chronic hepatitis B; CIMT: carotid artery intima-media thickness; C/T: computed tomography; DM: diabetes mellitus; FLI: fatty liver index; FMD: brachial flow-mediated vasodilatation; FBF: forearm blood flow; HOMA-IR: homeostatic model assessment for insulin resistance; LB: liver biopsy; MR: magnetic resonance; MS: metabolic syndrome; NAFL: nonalcoholic fatty liver; NAFLD: nonalcoholic fatty liver disease; NASH: nonalcoholic steatohepatitis; NMD: nitrate-mediated dilation; PAT: lower peripheral tonometry; PWV: carotid-femoral pulse wave velocity; PTX-3: pentraxin-3; U/S: ultrasound.

## Data Availability

Not applicable.

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
