# Peer review of "Endothelial Cell Dysfunction and Nonalcoholic Fatty Liver Disease (NAFLD): A Concise Review"

_cells, 2022, doi:10.3390/cells11162511_

Round 1
Reviewer 1 Report
The review entitled: “Endothelial Cell Dysfunction and Nonalcoholic Fatty Liver Disease (NAFLD)” by Nasiri-Ansari et al describe at length about how endothelial cell dysfunction is involved in the development, pathogenesis and progression of NAFLD. The review is comprehensive but similar to many other previous reviews in this area. The review will benefit from adding updated and new studies from the literature and also a few good figures.
Major Comments
· In the section, ‘LSECs in the regulation of inflammation in NAFLD’, the authors miss many of the reported mechanisms such as LSEC interaction with CD4/CD8 T cells, the CXC12-CXCR4 axis and also increased expression of TLRs, RUNX1 in these cells etc. These studies should be briefly discussed. The authors may also add a good figure explaining the role of LSECs in the regulation of inflammation in NAFLD.
· Also authors should add studies stating how LSECs interact with bone marrow-derived endothelial progenitor cells (EPCs) and facilitate inflammation and fibrosis in CCl4-induced toxin models.
· There are new studies in literature based on single cell sequencing that have reported novel markers in LSECs and potential targets of NASH. The authors should discuss them in a separate section.
· In the section, ‘Endothelial cell as a therapeutic target for the treatment of NAFLD’, studies using inhibitors to block the interaction of angiopoietin-2 and its receptor Tie2, Lipid nanoparticles targeting LSECs and its downstream effects etc are missing and need to be included appropriately. A good figure showing how LSEC targeting affects other liver cells may be added.
· In Figure 1, the authors have shown a cirrhotic or a NASH liver? Indeed this needs to be clearly written throughout the manuscript that how LSEC are functionally impaired in steatosis, NASH (with and without fibrosis) and Cirrhosis.
· The manuscript has many spelling and grammatical errors, which need to be corrected. A few of them have been pointed out below.
Minor Comments
· Please correct the word fenestrae in line no. 136.
· Line no. 156-160 is not clear.
· Line no 262, its LSECs instead of LCESs!
· Please use the abbreviation, ‘LSEC’ instead of liver sinusoidal endothelial cells throughout the manuscript after the first mention of the full form of LSECs.
· Page no. 8 line no. 377 correct the spelling of the word ‘characterization’
· Page no. 10, line no. 474 please correct “Of” (start with capital letter)
Author Response
August 7, 2022
We would like to thank the Reviewer for his/her thoughtful evaluation of our manuscript and the most welcome comments/suggestions. Accordingly, we have now revised our manuscript thoroughly to reflect these comments.
In the revised Text all changes/additions/modifications made in response to the Reviewer’s points (including linguistic revisions throughout the manuscript, reference renumbering within text, and newly added references and reference renumbering in the reference list) are marked up in red.
Please find below a point-by-point response to the issues raised by the Reviewer:
Reviewer 1
The review entitled: “Endothelial Cell Dysfunction and Nonalcoholic Fatty Liver Disease (NAFLD)” by Nasiri-Ansari et al describe at length about how endothelial cell dysfunction is involved in the development, pathogenesis and progression of NAFLD. The review is comprehensive but similar to many other previous reviews in this area. The review will benefit from adding updated and new studies from the literature and also a few good figures.
Response: We thank the Reviewer for his/her kind suggestions regarding our work.
Major Comments
In the section, ‘LSECs in the regulation of inflammation in NAFLD’, the authors miss many of the reported mechanisms such as LSEC interaction with CD4/CD8 T cells, the CXC12-CXCR4 axis and also increased expression of TLRs, RUNX1 in these cells etc. These studies should be briefly discussed. The authors may also add a good figure explaining the role of LSECs in the regulation of inflammation in NAFLD.
Response: We would like to thank the Reviewer for this comment which had a profound effect on improving our manuscript. The following parts and a new figure (Figure 2) have been added to the text:
CD4/CD8 T cells:
(Page 12/Lines 328-337)
LSECs and KCs are hepatic antigen-presenting cells, domestic to the liver sinusoidal lumen. Antigen presentation by LSECs to naive CD4+T and CD8+T cells is upregulated by inflammatory stimuli and induces T-cell differentiation towards regulatory phenotype (Treg) through activation of TGF-β and/or Notch-dependent signaling mechanisms [114-116]. In turn, Tregs induce fibrogenesis through increasing expression of both CD8+ and CD4+ T, as well as activation of Th17 cells [117]. Under physiological condition, LSECs-driven antigen presentation to CD8 T cells mediates naïve CD8+ T cells tolerance while in the presence of high levels of antigen this LSECs response is abrogated [114,115]. Additionally, LSECs are able to activate the naive CD4+ T cells and induce the expression of inflammatory cytokines by these cells. Study by Knolle et al. showed that antigen of purified murine (female 12-16-week-old BALB/c mice) LSECs can efficiently activate CD4+ T cells, as indicated by induced expression of IL-10, IL-4, and IFN-γ cytokines [115,118].
(Page 23/Lines 608-612)
ITGα4β7 regulates the binding of lymphocytes to its endothelial ligands, MAdCAM1 and VCAM-1 [177]. Treatment of diet-induced NASH mice model with ITGα4β7 antibody reduced liver inflammation, fibrosis, and metabolic dysfunction, as ITGα4β7 antibody reduced hepatic CD4+ T-cell homing [182]. During CCL4-induced liver fibrosis, naïve CD8+ T cells population were induced in WT mice leading to increase fibrosis and HSC activation the effect that was diminished after subtotal irradiation with a single total body dose of 700 cGy [183].
CXC12-CXCR4:
(Page 12/Lines 319-327)
CXCRs chemokine receptors and their CXC ligands (CXCLs) regulate migration and homing of inflammatory cells to the liver[109]. Aberrant expression of CXCR4 has been observed in NAFLD. Binding of CXCR4 to its ligand, CXCL12, regulates cell localization, chemotaxis, activation, migration, proliferation and differentiation [109,110]. CXCL12 also known as stromal cell-derived factor 1α (SDF-1α) is extensively produced by LSECs and induces HSCs migration during chronic liver injury [110]. Increased CXCR4 and CXCL12 protein levels along with aberrant CD4+ T cells response to CXCL12 have been observed during progression of NASH [110,111]. The hepatic recruitment of CD4+ T cells population is ease by LSECs through increased peri-vascular expression of CXCL12 and activation of CXCL12-CXCR4 dependent intracellular transport mechanisms [112]. CXCL12-CXCR4 axis activation induced HSCs proliferation and increased production of collagen I in CCL4 induced hepatic fibrosis mice model [113].
(Page 23/Lines 613-618 and Table 1)
A study by Mendt et al., revealed that LSECs conditional medium provoked the migration of BM progenitor lineage-negative (BM/Lin⁻) cells isolated from WT mice. This effect was abolished after incubation of cells with CXCR4 inhibitor known as AMD3100 [184]. Moreover, AMD3100 treatment reduced CD4+ T cells’ number and abolished the chemotactic effect of CXCL12 on CD4+ T in NASH mice model indicating that CXCR4 can represent a potential therapeutic target for NASH treatment [110]. On the contrary, gavage treatment of fibrotic mice with AMD070, another CXCR4 inhibitor, exerted a negligible effect on the liver fibrosis markers [185].
TLRs:
(Pages 12,13/Lines 338-350)
LSECs also express pattern recognition receptors such as stabilins and toll-like receptors (TLR 1-9). LSECs in response to TLR ligands stimulation, except for that of TLR5, activate inflammasome and inflammatory signaling [116]. In more details, LSECs produce either TNFα in response to TLR1, 2, 4, 6, 9 ligands, or TNFα, IL-6, and interferon (IFN) in response to TLR3 ligands [119]. Importantly, the above-mentioned LSECs response to TLRs through secretion of different type of cytokines is cell specific and was not observed in the KCs isolated from the same mice [120]. In NASH, the secretion of cytokines by LSECs leads to release of inflammatory mediators and therefore facilitate the progression of disease [121]. Some inflammatory-related signals in LSECs are mediated through the interaction between endocytosis receptors expressed on the LSECs and TLRs [2]. The TLR9 expressed by LSECs can uptake the bacterial DNA mimic CpG-oligonucleotides and activate endocytosis through scavenger receptor leading to secretion of inflammatory cytokines such as interleukin (IL)-1β and IL-6 [122]. In accordance, NAFLD activity scores, serum ALT levels, inflammatory cytokines expression, hepatic TGF-β and collagen I expression were reduced in TLR4 KO mice under MD diet as compared to WT mice fed MD [123]. Of note, study by Seki et al., demonstrated that fibrogenesis was significantly reduced after treament of TLR4-mutant mice with CCL4 or TAA [124].
(Page 23/Lines 619-621)
Treatment of primary LSECs isolated from mice with PA induced the expression of NOX-1 while this induction was repressed after incubation of cells with TAK-242 (TLR4-inhibitor) [80]. Of note, NOX-1 induces liver fibrosis through regulation of hepatocyte proliferation and ROS generation [80,186].
RUNX1:
(Page 13/Lines 351-360)
TLR4-mediated inflammatory signal is regulated by runt-related transcription factor 1 (RUNX1) [125]. RUNX1 is a lncRNA involved in regulation of oxidative-stress-induced angiogenesis and inflammation during NAFLD [125,126]. Increased expression of RUNX1 has been positively correlated with steatosis, fibrosis, and the degree of hepatic inflammation as well as NASH activity score, in NAFLD patients [125,127]. RUNX1 expression has been found to be up-regulated in both diet and CCL4 induced NASH mice models leading to HSCs activation and progression of NASH [128]. The expression of RUNX1 was increased in LSECs of mice under MD diet as compared to control. LSECs specific RUNX1 silencing of mice under MD diet resulted in reduced expression of ICAM-1, VCAM-1 and infiltration of immune cells in NASH [126]. RUNX1 elevated the expression of angiogenic, chemotactic factors and adhesion molecules in HUVEC cells, while these ECs properties were abolished after RUNX1 silencing, indicating the involvement of RUNX1 in enhancing inflammation and disease severity in NASH [127].
Figure 2: (Pages 14-15/Lines 375-395)
Figure 2. The LSECs’ anti-inflammatory and pro-inflammatory profile during progression of NAFLD.
At the early stage of NAFLD, LSECs display an anti-inflammatory function characterized by reduced expression of chemokines such as CCL2, CXCL10, and CXCL16 through MAPK signaling and induced expression of IL-10 by Th1 cells through activation of Notch signaling. Activation of Notch signaling manifests anti-inflammatory effects through induction of eNOS/sGC levels.
During the progression of NAFLD from simple steatosis to NASH and cirhossis, LSECs exhibit a pro-inflammatory phenotype mediated mostly through activation of NF-kB pathway. NF-kB regulates the expression of adhesion molecules (VCAM-1, ICAM-1, E-selectin, and VAP-1) as well as the secretion of pro-inflammatory cytokines (TNF-α, IL-1, and IL-6). The secretion of inflammatory mediators is also regulated by TLRs and NO bioavailability. Elevated expression of adhesion molecules leads to induced leukocytes recruitment and their translocation into the hepatic parenchyma. On the other hand, increased expression of inflammatory mediators along with LSECs dysfunction stimulates activation of KCs and leukocyte chemoattraction. Impaired LSECs’ autophagy observed during the progression of NAFLD also leads to upregulation of adhesion molecules and chemokines expression enhancing the inflammatory response. Reduced eNOS and increased iNOS contribute to the development of inflammation, activation of KCs and recruitment of bone marrow-derived macrophages.
Abbreviations: ICAM-1: intercellular adhesion molecule-1; IL-1: interleukin 1; IL-6: interleukin 6; LSECs: liver sinusoidal endothelial cells; MCP1: monocyte chemoattractant protein-1; NF-kB: nuclear factor kappa B; NO, nitric oxide; TNFa: tumor necrosis factor alpha; VAP-1: vascular adhesion protein1; VCAM-1: vascular cell adhesion molecule-1; BMMs:bone marrow-derived macrophages; TLR: Toll like Receptor; CXCL12:C-X-C Motif Chemokine Ligand 12; CXCR4: C-X-C chemokine receptor type 4; KC: Kupffer cells.
Also authors should add studies stating how LSECs interact with bone marrow-derived endothelial progenitor cells (EPCs) and facilitate inflammation and fibrosis in CCl4-induced toxin models.
Response: We thank the Reviewer for this thoughtful comment. The following part has been added to the text (Pages 10-11/Lines 270-292)
In parallel with LSECs capillarization, bone marrow derived endothelial progenitor cells (BM-EPCs) are increased du- ring chronic liver diseases [92,93]. LSECs capillarization induces hypoxia due to elevated resistance to blood flow and oxygen delivery from the sinusoids to the parenchyma, leading to increased HSCs activation and expression of VEGF, angiopoietins and their receptors [93-96]. A paracrine crosstalk between BM-EPCs with LSECs via VEGF and PDGF has previously been reported. In particular, a study by Kaur et al. showed that the interaction between circulating BM-EPCs and resident LSECs enhances angiogenic functions via induction of paracrine mediators such as VEGF and PDGF-BB. The same study reported induced circulating BM-EPCs levels in patients with cirrhosis as compared to controls [97]. Another study from the same research group indicated that there is a substantial positive correlation between abundance of BM-EPCs and fibrosis during the early stage of the liver injury in mice (after 4 weeks of CCL4 treatment) while after 8 weeks of CCL4 treatment EPC levels returned back to basal levels, most likely due to lack of demand for hepatic tissue regenerating, indicating thus the potential role of BM-EPs in the early stage of liver fibrosis [93].
Another study by Liu et al., demonstrated that the intraperitoneal injection of cultured EPCs to rats under CCL4 treatment for 8 weeks exerts hepato-protective effect as indicated by reduced ALT and AST levels and decreased liver fibrogenesis [98,99]. In line with these findings, injection of cells derived from high-density (HD) culture of rat’s bone marrow cells enriched in BM-EPCs to the CCL4 treated rats improved both biochemical and fibrotic markers of liver injury after 4 weeks-post transplantation. Importantly, the transplanted EPCs were not differentiated into either hepatocytes or endothelial cells, confirming that the BM-EPCs exert these beneficial effects most likely through acting on surrounding cells rather than their direct interaction [93,100]. Increased number of BM-EPCs in NAFLD patients was considered as compensatory mechanism against endothelial injury observed during the NAFLD progression and it was proportionally associated with the degree of liver steatosis [96,101]. On the contrary, reduced number and function of circulating BM-EPCs in patients with NAFLD was reported by Chiang et al. This study revealed that the circulating BM-EPCs population can be used as an independent reverse predictor of NAFLD [102].
There are new studies in literature based on single cell sequencing that have reported novel markers in LSECs and potential targets of NASH. The authors should discuss them in a separate section.
Response: We thank the reviewer for this comment. The following section has been added to the text (Pages 16-17/Lines 433-466)
3.6. Novel markers of LSECs dysfunction
While various markers such as CD32b, CLEC4G, LYVE1 and STAB2 have been emerged for the detection of LSECs in healthy livers, electron microscopy remains the only gold standard for identification of damaged LSECs as it can detect the loss of basement membrane and fenestrae [142].
Recently, single-cell transcriptomic (scRNAseq) analysis in both healthy and diseased human and mouse livers has identified heterogeneity within LSECs population and several biomarkers to assess disease development [142,143].
The transcriptome analysis of more than 100,000 single human cells revealed seven distinct endothelial subpopulations that inhabit in the fibrotic niche. These endothelial cells express both ACKR1+ and PLVAP+ which are restricted to cirrhotic liver tissue and induce the transmigration of leucocytes. Importantly, ACKR1 knockdown attenuated leucocytes recruitment by cirrhotic endothelial cells. Metagene signature analysis revealed that the expression of pro-fibrogenic genes such as PDGFD, PDGFB, LOX, LOXL2 in the scared associated endothelial cells was associated with extracellular matrix rearrangement and increased hepatic fibrillar collagens [144]. While the expression of LSECs specific scavenger receptors including STAB2, CLEC4G, CD209, MRC1, and CD32B, as well as receptors involved in VEGF-induced angiogenesis signaling such as KDR and NRP1 have been defined as a signature of healthy LSECs [142], a study by Verhulst and colleagues showed that STAB2 and CLEC4G are reduced during chronic liver diseases [142]. Transcriptomics revealed that the interaction between LSECs and chemokines was disrupted in deficient for STAB1 (Stab1KO) and STAB2 (Stab1KO) mice due to reduced expression of adhesion molecules and other molecules involved in cytokine-cytokine receptor interaction [145]. Strong expression of both TIMP1 and TIMP2 in LSECs was observed in single cell analysis of human livers of both chronic and acute liver injury [142]. On the contrary, a study by Xiong et al., found no significant changes in the expression of both TIMP1 and TIMP2 while they detected an abundant expression of Fcgr2b and Gpr182 by scRNAseq analysis [146]. The expression of endothelial cellular markers of lipid accumulation Cxcl9 and BODIPY was strongly elevated in LSECs isolated from trans-fat containing amylin liver NASH (AMLN-diet) induced NASH mouse model. Microarray dataset analysis of published data containing samples from 24 healthy, 20 NAFLD, and 19 NASH patients (GEO: GSE89632 [147] ) by Xiong et al., showed that similar to NASH mouse, during human NASH pathogenesis there is a hepatic transcript abundance of CXCL9 and FABP4 while the expression of BMP2, NRP1, and VEGFA was reduced in the hepatic tissue of patients with NAFLD and NASH [146].
Apart from the aforementioned markers of LSECs function/dysfunction, the expression of FABP4, Fatty Acid-Binding Protein 5 (FAPB5), Von Willebrand Factor (VWF), Von Willebrand Factor A Domain Containing 1 (VWA1) and CD31 has been found to be upregulated in LSECs during liver disease [49,142,148]. A single cell RNAseq analysis by Verhulst and colleagues revealed the up regulation of FABP4/5 and VWF/a1 as a signature of damaged human LSECs [142]. The expression of FABP4 has been found to be elevated during liver fibrosis. FABP4 promotes LSECs capilarization and therefore plays a crucial role during the onset and progression of liver fibrosis in mice [149]. The expression of VWF was not detected in healthy livers while it was increased in LSECs fibrotic livers obtained from CCL4 treated mice and rats as well as NASH rats with or without cirrhosis [150,151].
In the section, ‘Endothelial cell as a therapeutic target for the treatment of NAFLD’, studies using inhibitors to block the interaction of angiopoietin-2 and its receptor Tie2, Lipid nanoparticles targeting LSECs and its downstream effects etc are missing and need to be included appropriately. A good figure showing how LSEC targeting affects other liver cells may be added.
Response: We thank the Reviewer for this comment. The following section has been added to the text and Table 1 has been updated.
Ang-2/Tie2 interaction
(Page 22/Lines 586-589)
Hepatic angiogenesis promotes chronic inflammation in NASH, while inhibition of angiogenesis has been shown to improve hepatic inflammation. Hepatic angiogenesis is regulated by both vascular endothelial growth factor (VEGF/VEGFR) and angiopoietin /tyrosine kinase with immunoglobulin-like and EGF-like domains 2 (Ang/Tie2) pathways [177,178].
(Pages 22-23/Lines 600-607)
Lefere and colleagues showed that Inhibition of Ang-2/Tie2 interaction by peptibody L1-10 alleviates ballooning, fibrosis, as well as hepatic inflammation, angiogenesis and microvascular architecture distortion in MCD diet-induced NASH and streptozotocin-western diet-induced NASH mice models [178]. The hepato-protective effect of L1-10 therapy seems to be, at least in part, through its effects on LSECs since L1-10 reduced the expression of VCAM-1, ICAM-1 and MCP1 in LSECs isolated from mice fed a methionine- and choline-deficient diet. Interestingly, blocking the Ang-2 signaling in the streptozotocin-western diet NASH mice model resulted in reduced VCAM-1 expression, reversed NASH and ameliorated the HCC progression. Additionally, in vitro treatment of LPS-stimulated LSECs with L1-10 led to decreased expression of inflammatory markers [177,178].
Nano particles (Pages 23-24/Lines 622-650)
Currently, nanoparticles (NPs) drug delivery methods have attracted tremendous attention as a therapeutic approach for the treatment of NAFLD. NPs represent an opportunity to achieve sophisticated targeting therapy due to their size and surface characteristics and their ability to protect drug degradation and control the drug cellular uptake at the desired tissues such as the liver [187]. Due to the lack of basal lamina and the size of liver sinusoidal fenestrae (50–200 nm), LSECs provide a mesh-like structure contributing to the entrapment of NPs in the liver. This structure facilitates the accumulation of high concentration of NPs in the Disse space and their distribution to other liver cells [188].
After systemic administration, NPs with a size greater than 6 nm accumulate in the liver therefore enterohepatic micro-circulation plays a crucial role in delivering orally administered NPs to the liver [189]. Recently, apolipoprotein B (ApoB) sequence has been used to decorate nanoparticles, given that ApoB is a ligand for both scavenger receptors stabilin-1 and 2 expressed by LSECs [138].
Hyaluronic acid (HA) micelles targeting HA receptor can be used for targeting both LSECs and HSCs. HA micelles particles carrying losartan were shown to be an effective NP delivery system that ameliorates the advanced liver fibrosis in a C3H/HeN mouse model, as demonstrated by reduced ALT and AST serum and decreased hepatic alpha smooth muscle actin (α-SMA) expression [190]. Moreover, lipid NPs carrying procollagen α I(I) siRNA remarkably reduced the total hepatic collagen content leading to alleviation of NASH progression and accelerating regression of hepatic fibrosis in CCL4-induced NASH in Balb/c mice [191]. In another attempt, CXCR4-targeted lipid-based NPs carrying VEGF siRNA was used for the treatment of liver fibrosis and hepatic cellular carcinoma; combination therapy using both AMD-NPs (CXCR4 inhibitor) and VEGF siRNA abolished the infiltration of tumor-associated macrophages and reduced HCC tumor growth [192].
Apart from the above-mentioned, use of quantum dots (QDs) nanoparticles (1–20 nm) for treatment of NAFLD has also been emerged as an area or great interest [193]. Intravenous injection of mercaptosuccinic acid (MSA)-capped cadmium telluride/cadmium sulfide (CdTe/CdS/QDs) showed that negatively charged QDs were selectively taken up by sinusoidal cells (KCs and LSECs) in rat liver, indicating that CdTe/CdS/QDs can be used for drug delivery to the LSECs [194]. Furthermore, Zn-labelled CdSelenide/CdS/ZnS QDs have also been found in the KCs and LSECs, 2h after intravenous injection of polymer-coated Qdots indicating their efficiency in drug delivery by the sinusoidal cells [195]. Besides that, via usage of nanoparticles carrying Honokiol or adenovirus with the endothelial cell specific arginylglycylaspartic acid-roundabout guidance receptor 4 (RGD-ROBO4) ERK1/2 activation was promoted leading to induced liver regeneration and reduced hepatic fibrosis, due to high degree of LSEC specificity [196].
In Figure 1, the authors have shown a cirrhotic or a NASH liver? Indeed this needs to be clearly written throughout the manuscript that how LSEC are functionally impaired in steatosis, NASH (with and without fibrosis) and Cirrhosis.
Response: We thank the Reviewer for this comment. The following section has been added to the text in order to address LSECs dysfunction during different stages of NAFLD. However, the exact mechanism of LSECs’ impairment during each stage of NAFLD has not been fully understood.
(Pages 8-9/Lines 227-232)
Circulating lipids seem to induce oxidative stress in LSECs, while this oxidative stress contributes to hepatocyte injury resulting in NASH [79]. Indeed, treatment of primary murine cultured LSECs with palmitic acid (PA) upregulated the expression of the NOX1 isoform of NADPH oxidase, an enzyme implicated in ROS production [80]. Furthermore, NOX1 was also upregulated in the liver of NASH patients and mice fed a high-fat and high-cholesterol (HFC) diet for 8 weeks, while mice deficient in NOX1 displayed decreased levels of serum alanine aminotransferase (ALT) and hepatic cleaved caspase-3 compared to wild-type littermates when fed the HFC diet [80].
(Page 9/Lines 248-257)
Capillarization of LSECs occurs in a very early phase of NAFLD, even prior to steatosis establishment [79]. Indeed, LSEC defenestration begins after 1 week of choline-deficient, L-amino acid-defined (CDAA) diet administration in mice [82], and LSEC morphology is damaged after 3 weeks of HFD feeding in rats [83]. Capillarization then leads to liver steatosis, as shown in mice deficient in plasmalemma vesicle-associated protein (PLVAP), an endothelial-specific, integral membrane glycoprotein that has been identified to be a component of endothelial fenestrae [81]. The LSECs of these mice exhibit a significant reduction in the number of fenestrations, associated with a decrease in the transport of macromolecules from the sinusoidal lumen into Disse space, which leads to the development of extensive multivesicular steatosis, followed by steatohepatitis and fibrosis [81]. LSECs capillarization occurs before KCs activation and is permissive of hepatic stellate cell activation, and progression of inflammation and fibrosis [8,79,84].
(Page 11/Lines 302-306)
When NAFLD progresses to NASH, the LSECs display a pro-inflammatory phenotype characterized by the surface overexpression of adhesion molecules such as ICAM-1, VCAM-1 and VAP-1 (AOC3) and the production of pro-inflammatory molecules, including TNF-α, IL-6, IL-1 and MCP1 (CCL2), as it has been observed in experiments in mouse models of NASH [95,104]. The monocytes that adhered to the LSECs and trapped in the sinusoids play pivotal role in the initiation and progression of NAFLD [8,49,54].
(Page 16/Lines 422-431)
3.5. LSECs in NAFLD-related HCC
LSECs have been also implicated in the progression of NAFLD to HCC, although availabe data are scarse. In 2009, Milner et al. reported that the adipokine fatty acid–binding protein 4 (FABP4) was elevated in NAFLD patients without HCC versus healthy controls, distinguishing steatohepatitis from simple steatosis and predicting liver inflammation and fibrosis [140]. Recently, it was demonstrated that FABP4 was overexpressed in human HCC samples from patients with metabolic syndrome and this expression was mainly detected in peritumoral endothelial cells [141]. Interestingly, though FABP4 is not expressed by LSECs under basal conditions, the exposure of these cells to conditions mimicking NAFLD (high concentrations of glucose, insulin, and VEGFA) led to a significant release of FABP4 protein and FABP4 increased cell viability and proliferation of hepatocytes, leading to the conclusion that FABP4 exerts pro-oncogenic effects [141].
We have also changed the legend to Figure 1 accordingly (Pages 5-6/Lines 129-154)
Figure 1. The role of endothelial cells in NAFLD pathogenesis and the interplay between CVD and NAFLD.
LSECs are located at the interface between blood stream and liver parenchyma. LSECs regulate blood flow in response to shear stress mainly through increased NO synthesis and bioavailability, as well as through ET-1 reduction which are both mediated by KLF2. LSECs also regulate the activation of KCs and HSCs during NASH progression. The expression of SRs, MR, and FcγRIIb2 endows LSECs with high endocytic capacity; of note, reduced endocytic capicity of LSECs precedes fibrosis in NAFLD. The increased expression of adhesion molecules during the early stage of NAFLD enhances the recruitment of monocytes to the inflamed endothelium leading to the activation of inflammatory response in NASH. Impaired autophagy has been associated with the development of steatosis and fibrosis, through –among others- up-regulation of adhesion molecules and pro-inflammatory mediators during the progression of the disease. Moreover, hepatocyte-derived EVs contribute to the formation of inflammatory foci by recruitment of macrophages into the hepatic sinusoids. Both HSCs- and LSECs-drived EVs play a crucial role in maintenance of the balance between extracellular matrix production and degradation and the consequent progression towards regeneration of hepatic cells or fibrosis. The anti-inflammatory features of LSECs observed during the early stage of NAFLD developments are attributable –among others- to decreased CCL, CXCL expression through MAPK signaling activation and increased secretion of IL-10 by Th1 cells. NAFLD is strongly related to vascular endothelial dysfunction and consequence atherosclerosis. Overexpression of inflammatory mediators, elevated insulin resistance and oxidative stress are key-players in this interrelation. Increased levels of inflammatory molecules such as circulating fetuin-A, ADMA, cRP and SeP have been associated with elevated risk of CVDs in NAFLD patients and vice versa : the low-grade inflammatory millieu of atherosclerosis could promote the progression of NAFLD.
This image was derived from the free medical site http: //smart.servier.com/ (accessed on June 2022) by Servier, licensed under a Creative Commons Attribution 3.0 Unported licence.
Abbreviations: CCL: C-C motif chemokine ligand; CCR: C-C motif chemokine receptor; CXCL: C-X-C motif chemokine ligand; NO; nitric oxide; ICAM-1: intercellular adhesion molecule-1; LSEC: liver sinusoidal endothelial cell; TNF-α: tumor necrosis factor-α; VAP-1: vascular adhesion protein-1; VCAM-1: vascular cell adhesion molecule-1.; N: neutrophils; ROS: reactive oxygen species; VEGF: vascular endothelial growth factor; VEGFR: vascular endothelial growth factor receptor; EV: extracellular vesicles; KLF2: Kruppel-like factor 2; M: monocytes; KC: Kupffer cells; IR: insulin resistance; HSCs: hepatic stellate cells; SR: scavenger receptor; MR: Mannose receptor; FcγRIIb2: Fc gamma receptor IIb.
The manuscript has many spelling and grammatical errors, which need to be corrected. A few of them have been pointed out below.
Response: We have corrected the spelling and grammar errors throughout the manuscript.
Minor Comments
Please correct the word fenestrae in line no. 136.
Response: We have the indicated word.
Line no. 156-160 is not clear.
Response: We have corrected this sentence to make it clear.
Line no 262, its LSECs instead of LCESs!
Response: We have corrected the misspelling.
Please use the abbreviation, ‘LSEC’ instead of liver sinusoidal endothelial cells throughout the manuscript after the first mention of the full form of LSECs.
Response: Corrected throughout the manuscript according to the Reviewer’s suggestion.
Page no. 8 line no. 377 correct the spelling of the word ‘characterization’ (line 316)
Response: We have corrected the spelling.
Page no. 10, line no. 474 please correct “Of” (start with capital letter)
Response: Corrected.
Trusting that we have adequately addressed the Reviewer’s concerns, we would like to thank him/her for his/her help in improving significantly our work.
With kind regards,
Prof. Athanasios G. Papavassiliou, MD, PhD
Prof. Eva Kassi, MD, PhD
Corresponding authors
Reviewer 2 Report
Major comments
- The authors should provide a deeper explanation on LSECs communication with T cells as they name it in the text. Also, please provide detailed inflammatory cytokines in Figure 1, or add another one more specific for the immune profile.
- As single cell studies are increasing in number, and a lot of information is available, please discuss these results for LSECs in NAFLD with the most relevant published information.
- I would recommend an improvement of the main figure, figure 1, with a “before” and “after” representation of the LSECs situation in the context of NAFLD, or during the progression of the pathology.
Minor comments
The paper is a very nice comprehensive review on the field of NAFLD. I suggest a deeper review on English spelling an an improvement in the use of English expressions or abbreviation definition.
As an example, please aware of:
Line 58: microenvironment
Line 120: correct the acronym definition
Line 136: fenestrae
Line 158: figure 1 legend- quite different spelling mistakes
Line 262: Title error
Line 415: cirrhosis
Line 474: "of note"
Table 1: line with reference 125= adhesion
Author Response
August 7, 2022
We would like to thank the Reviewer for his/her thoughtful evaluation of our manuscript and the most welcome comments/suggestions. Accordingly, we have now revised our manuscript thoroughly to reflect these comments.
In the revised Text all changes/additions/modifications made in response to the Reviewer’s points (including linguistic revisions throughout the manuscript, reference renumbering within text, and newly added references and reference renumbering in the reference list) are marked up in red.
Please find below a point-by-point response to the issues raised by the Reviewer:
Reviewer 2
The authors should provide a deeper explanation on LSECs communication with T cells as they name it in the text. Also, please provide detailed inflammatory cytokines in Figure 1, or add another one more specific for the immune profile.
Response: We would like to thank the Reviewer for this comment. The following parts and a new figure (Figure 2) have been added to the text:
(Page 12/Lines 328-337)
LSECs and KCs are hepatic antigen-presenting cells, domestic to the liver sinusoidal lumen. Antigen presentation by LSECs to naive CD4+T and CD8+T cells is upregulated by inflammatory stimuli and induces T-cell differentiation towards regulatory phenotype (Treg) through activation of TGF-β and/or Notch-dependent signaling mechanisms [114-116]. In turn, Tregs induce fibrogenesis through increasing expression of both CD8+ and CD4+ T, as well as activation of Th17 cells [117]. Under physiological condition, LSECs-driven antigen presentation to CD8 T cells mediates naïve CD8+ T cells tolerance while in the presence of high levels of antigen this LSECs response is abrogated [114,115]. Additionally, LSECs are able to activate the naive CD4+ T cells and induce the expression of inflammatory cytokines by these cells. Study by Knolle et al. showed that antigen of purified murine (female 12-16-week-old BALB/c mice) LSECs can efficiently activate CD4+ T cells, as indicated by induced expression of IL-10, IL-4, and IFN-γ cytokines [115,118].
(Page 23/Lines 608-612)
ITGα4β7 regulates the binding of lymphocytes to its endothelial ligands, MAdCAM1 and VCAM-1 [177]. Treatment of diet-induced NASH mice model with ITGα4β7 antibody reduced liver inflammation, fibrosis, and metabolic dysfunction, as ITGα4β7 antibody reduced hepatic CD4+ T-cell homing [182]. During CCL4-induced liver fibrosis, naïve CD8+ T cells population were induced in WT mice leading to increase fibrosis and HSC activation the effect that was diminished after subtotal irradiation with a single total body dose of 700 cGy [183].
Figure 2: (Pages 14-15/Lines 375-395)
Figure 2. The LSECs’ anti-inflammatory and pro-inflammatory profile during progression of NAFLD.
At the early stage of NAFLD, LSECs display an anti-inflammatory function characterized by reduced expression of chemokines such as CCL2, CXCL10, and CXCL16 through MAPK signalling and induced expression of IL-10 by Th1 cells through activation of Notch signalling. Activation of Notch signaling manifests anti-inflammatory effects through induction of eNOS/sGC levels.
During the progression of NAFLD from simple steatosis to NASH and cirhossis, LSECs exhibit a pro-inflammatory phenotype mediated mostly through activation of NF-kB pathway. NF-kB regulates the expression of adhesion molecules (VCAM-1, ICAM-1, E-selectin, and VAP-1) as well as the secretion of pro-inflammatory cytokines (TNF-α, IL-1, and IL-6). The secretion of inflammatory mediators is also regulated by TLRs and NO bioavailability. Elevated expression of adhesion molecules leads to induced leukocytes recruitment and their translocation into the hepatic parenchyma. On the other hand, increased expression of inflammatory mediators along with LSECs dysfunction stimulates activation of KCs and leukocyte chemoattraction. Impaired LSECs’ autophagy observed during the progression of NAFLD also leads to upregulation of adhesion molecules and chemokines expression enhancing the inflammatory response. Reduced eNOS and increased iNOS contribute to the development of inflammation, activation of KCs and recruitment of bone marrow-derived macrophages.
Abbreviations: ICAM-1: intercellular adhesion molecule-1; IL-1: interleukin 1; IL-6: interleukin 6; LSECs: liver sinusoidal endothelial cells; MCP1: monocyte chemoattractant protein-1; NF-kB: nuclear factor kappa B; NO, nitric oxide; TNFa: tumor necrosis factor alpha; VAP-1: vascular adhesion protein1; VCAM-1: vascular cell adhesion molecule-1; BMMs:bone marrow-derived macrophages; TLR: Toll like Receptor; CXCL12:C-X-C Motif Chemokine Ligand 12; CXCR4: C-X-C chemokine receptor type 4; KC: Kupffer cells.
As single cell studies are increasing in number, and a lot of information is available, please discuss these results for LSECs in NAFLD with the most relevant published information.
Response: We would like to thank the Reviewer for this thoughtful comment. The following section has been added to the main text (Pages 16-17/Lines 433-466)
3.6. Novel markers of LSECs dysfunction
While various markers such as CD32b, CLEC4G, LYVE1 and STAB2 have been emerged for the detection of LSECs in healthy livers, electron microscopy remains the only gold standard for identification of damaged LSECs as it can detect the loss of basement membrane and fenestrae [142].
Recently, single-cell transcriptomic (scRNAseq) analysis in both healthy and diseased human and mouse livers has identified heterogeneity within LSECs population and several biomarkers to assess disease development [142,143].
The transcriptome analysis of more than 100,000 single human cells revealed seven distinct endothelial subpopulations that inhabit in the fibrotic niche. These endothelial cells express both ACKR1+ and PLVAP+ which are restricted to cirrhotic liver tissue and induce the transmigration of leucocytes. Importantly, ACKR1 knockdown attenuated leucocytes recruitment by cirrhotic endothelial cells. Metagene signature analysis revealed that the expression of pro-fibrogenic genes such as PDGFD, PDGFB, LOX, LOXL2 in the scared associated endothelial cells was associated with extracellular matrix rearrangement and increased hepatic fibrillar collagens [144]. While the expression of LSECs specific scavenger receptors including STAB2, CLEC4G, CD209, MRC1, and CD32B, as well as receptors involved in VEGF-induced angiogenesis signalling such as KDR and NRP1 have been defined as a signature of healthy LSECs [142], a study by Verhulst and colleagues showed that STAB2 and CLEC4G are reduced during chronic liver diseases [142]. Transcriptomics revealed that the interaction between LSECs and chemokines was disrupted in deficient for STAB1 (Stab1KO) and STAB2 (Stab1KO) mice due to reduced expression of adhesion molecules and other molecules involved in cytokine-cytokine receptor interaction [145]. Strong expression of both TIMP1 and TIMP2 in LSECs was observed in single cell analysis of human livers of both chronic and acute liver injury [142]. On the contrary, a study by Xiong et al., found no significant changes in the expression of both TIMP1 and TIMP2 while they detected an abundant expression of Fcgr2b and Gpr182 by scRNAseq analysis [146]. The expression of endothelial cellular markers of lipid accumulation Cxcl9 and BODIPY was strongly elevated in LSECs isolated from trans-fat containing amylin liver NASH (AMLN-diet) induced NASH mouse model. Microarray dataset analysis of published data containing samples from 24 healthy, 20 NAFLD, and 19 NASH patients (GEO: GSE89632 [147] ) by Xiong et al., showed that similar to NASH mouse, during human NASH pathogenesis there is a hepatic transcript abundance of CXCL9 and FABP4 while the expression of BMP2, NRP1, and VEGFA was reduced in the hepatic tissue of patients with NAFLD and NASH [146].
Apart from the aforementioned markers of LSECs function/dysfunction, the expression of FABP4, Fatty Acid-Binding Protein 5 (FAPB5), Von Willebrand Factor (VWF), Von Willebrand Factor A Domain Containing 1 (VWA1) and CD31 has been found to be upregulated in LSECs during liver disease [49,142,148]. A single cell RNAseq analysis by Verhulst and colleagues revealed the up regulation of FABP4/5 and VWF/a1 as a signature of damaged human LSECs [142]. The expression of FABP4 has been found to be elevated during liver fibrosis. FABP4 promotes LSECs capilarization and therefore plays a crucial role during the onset and progression of liver fibrosis in mice [149]. The expression of VWF was not detected in healthy livers while it was increased in LSECs fibrotic livers obtained from CCL4 treated mice and rats as well as NASH rats with or without cirrhosis [150,151].
I would recommend an improvement of the main figure, figure 1, with a “before” and “after” representation of the LSECs situation in the context of NAFLD, or during the progression of the pathology.
Response: We thank the Reviewer for this comment. Due to abundant amount of information presented in Figure 1, we weren’t able to add more details to this figure. Adding new information would make this figure too busy to be followed by the reader. However, according to the Reviewer’s suggestion we have now provided more detailed information in the legend to Figure 1 and we have also added a new figure (Figure 2) to the text.
We have changed the legend to Figure 1 as follows (Pages 5-6/Lines 129-154)
Figure 1. The role of endothelial cells in NAFLD pathogenesis and the interplay between CVD and NAFLD.
LSECs are located at the interface between blood stream and liver parenchyma. LSECs regulate blood flow in response to shear stress mainly through increased NO synthesis and bioavailability, as well as through ET-1 reduction which are both mediated by KLF2. LSECs also regulate the activation of KCs and HSCs during NASH progression. The expression of SRs, MR, and FcγRIIb2 endows LSECs with high endocytic capacity; of note, reduced endocytic capicity of LSECs precedes fibrosis in NAFLD. The increased expression of adhesion molecules during the early stage of NAFLD enhances the recruitment of monocytes to the inflamed endothelium leading to the activation of inflammatory response in NASH. Impaired autophagy has been associated with the development of steatosis and fibrosis, through –among others- up-regulation of adhesion molecules and pro-inflammatory mediators during the progression of the disease. Moreover, hepatocyte-derived EVs contribute to the formation of inflammatory foci by recruitment of macrophages into the hepatic sinusoids. Both HSCs- and LSECs-drived EVs play a crucial role in maintenance of the balance between extracellular matrix production and degradation and the consequent progression towards regeneration of hepatic cells or fibrosis. The anti-inflammatory features of LSECs observed during the early stage of NAFLD developments are attributable –among others- to decreased CCL, CXCL expression through MAPK signaling activation and increased secretion of IL-10 by Th1 cells. NAFLD is strongly related to vascular endothelial dysfunction and consequence atherosclerosis. Overexpression of inflammatory mediators, elevated insulin resistance and oxidative stress are key-players in this interrelation. Increased levels of inflammatory molecules such as circulating fetuin-A, ADMA, cRP and SeP have been associated with elevated risk of CVDs in NAFLD patients and vice versa : the low-grade inflammatory millieu of atherosclerosis could promote the progression of NAFLD.
This image was derived from the free medical site http: //smart.servier.com/ (accessed on June 2022) by Servier, licensed under a Creative Commons Attribution 3.0 Unported licence.
Abbreviations: CCL: C-C motif chemokine ligand; CCR: C-C motif chemokine receptor; CXCL: C-X-C motif chemokine ligand; NO; nitric oxide; ICAM-1: intercellular adhesion molecule-1; LSEC: liver sinusoidal endothelial cell; TNF-α: tumor necrosis factor-α; VAP-1: vascular adhesion protein-1; VCAM-1: vascular cell adhesion molecule-1.; N: neutrophils; ROS: reactive oxygen species; VEGF: vascular endothelial growth factor; VEGFR: vascular endothelial growth factor receptor; EV: extracellular vesicles; KLF2: Kruppel-like factor 2; M: monocytes; KC: Kupffer cells; IR: insulin resistance; HSCs: hepatic stellate cells; SR: scavenger receptor; MR: Mannose receptor; FcγRIIb2: Fc gamma receptor IIb.
The paper is a very nice comprehensive review on the field of NAFLD. I suggest a deeper review on English spelling and an improvement in the use of English expressions or abbreviation definition.
Response: We thank the Reviewer for his/her positive evaluation of our work. Following the Reviewer’s suggestion, an extensive review on English spelling and an improvement in the use of English expressions and abbreviation definition has been performed in the revised manuscript.
As an example, please aware of:
Line 58: microenvironment
Response: We have corrected the misspelling.
Line 120: correct the acronym definition
Response: We have corrected the acronym definition.
Line 136: fenestrae
Response: We have corrected the misspelling.
Line 158: figure 1 legend- quite different spelling mistakes
Response: We have corrected the spelling mistakes.
Line 262: Title error
Response: We have corrected the title error.
Line 415: cirrhosis
Response: We have corrected the misspelling.
Line 474: "of note"
Response: Corrected.
Table 1: line with reference 125= adhesion
Response: We have corrected the misspelling.
Trusting that we have adequately addressed the Reviewer’s concerns, we would like to thank him/her for his/her help in improving significantly our work.
With kind regards,
Prof. Athanasios G. Papavassiliou, MD, PhD
Prof. Eva Kassi, MD, PhD
Corresponding authors

Round 2
Reviewer 1 Report
The manuscript has been revised adequately and the authors have addressed all concerns.